# Glaucoma-associated Optineurin mutations increase transcellular degradation of mitochondria in a vertebrate optic nerve

Yaeram Jeong[1], Chung-ha O Davis[2], Aaron M Muscarella[3], Hector H Navarro[1], Viraj Deshpande[1], Lucy G Moore[1], Keun-Young Kim[3], Mark H Ellisman[3], Nicholas Marsh-Armstrong[1]*

[1]Department of Ophthalmology and Vision Science, University of California Davis School of Medicine, Davis, United States; [2]Neurosciences Program, Stanford University, Stanford, United States; [3]National Center for Microscopy and Imaging Research, Center for Research in Biological Systems, Department of Neurosciences, University of California San Diego School of Medicine, San Diego, United States

*For correspondence:
nmarsharmstrong@ucdavis.edu

Competing interest: The authors declare that no competing interests exist.

**Abstract** We previously described a process whereby mitochondria shed by retinal ganglion cell (RGC) axons are transferred to and degraded by surrounding astrocytes in the optic nerve head of mice. Since the mitophagy receptor Optineurin (OPTN) is one of few large-effect glaucoma genes and axonal damage occurs at the optic nerve head in glaucoma, here we explored whether OPTN mutations perturb the transcellular degradation of mitochondria. Live imaging of *Xenopus laevis* optic nerves revealed that diverse human mutant but not wild-type OPTN increase stationary mitochondria and mitophagy machinery and their co-localization within, and in the case of the glaucoma-associated OPTN mutations also outside of, RGC axons. These extra-axonal mitochondria are degraded by astrocytes. Our studies demonstrate that expression of OPTN carrying a glaucoma-associated mutation results in increased transcellular degradation of axonal mitochondria.

## Editor's evaluation

This work describes the effect of Optineurin (OPTN) mutations in the transcellular degradation of retinal ganglion cell mitochondria by astrocytes in the Optic Nerve, a process termed, "transcellular degradation of mitochondria". The authors perform compelling live imaging studies of the *Xenopus laevis* optic nerve to track neuronal mitochondrial movement and expulsion in an intact nervous system. These important findings demonstrate that Optineurin mutations that are associated with disease increase the stationary pool of mitochondria resulting in increased rates of transcellular degradation.

## Introduction

Mitochondria carry out diverse functions including being the major source of energy production (*Attwell and Laughlin, 2001*), and thus the proper quality control of mitochondria is essential for viability (*Chen and Chan, 2009*; *Schon and Przedborski, 2011*; *Rugarli and Langer, 2012*). Mitophagy entails either the constitutive or induced degradation of damaged or unnecessary mitochondria using highly conserved autophagic pathways that are also used to degrade other organelles, aggregates, and pathogens (*Ashrafi and Schwarz, 2013*; *Ding and Yin, 2012*; *Youle and Narendra,*

*2011*). Mitochondria are often targeted for degradation by a process regulated by a sequence of phosphorylation and ubiquitination steps regulated by two genes that have been linked to Parkinson's disease, PTEN-induced putative kinase 1 (PINK1) and Parkin (*Kitada et al., 1998*; *Narendra et al., 2008*). Optineurin (OPTN) is one of the principal mitophagy receptors that recognizes these ubiquitinated mitochondria (*Wong and Holzbaur, 2014a*; *Heo et al., 2015*; *Lazarou et al., 2015*), others being Nuclear dot protein 52 kDa (NDP52), neighbor of BRCA1 gene 1 (NBR1) and Sequestosome-1 (SQSTM1 or p62) (*Rogov et al., 2014*). Once OPTN is recruited to the ubiquitinated mitochondria, these mitochondria can then associate with microtubule-associated protein 1A/1B-light chain 3 (LC3) on nascent autophagosomes through the LC3 interacting region of OPTN, leading to full engulfment of the tagged organelles inside of the autophagosomes, followed by lysosomal fusion that enables the subsequent degradation of the contained mitochondria (*Stolz et al., 2014*).

Although mitochondria are primarily produced and degraded in or near the cell soma (*Burke et al., 1988*; *Saxton and Hollenbeck, 2012*), many neuronal mitochondria also reside in distal processes far from the cell body (*Nafstad and Blackstad, 1966*), a situation that is the most extreme in the axons of long projection neurons such as retinal ganglion cells (RGCs) (*Yu et al., 2013*). To date, most studies have focused on mitochondria quality control mechanisms in the soma of cells, and much less is known about mitochondria quality control in axons. Since effective clearance of damaged organelles in long axons is likely essential for their proper function, it has been a long-standing question whether adequate mitochondria quality control, more specifically efficient removal of dysfunctional mitochondria, can occur at these long processes (*Lu, 2014*). There are studies that showed that distal axonal damaged mitochondria, or alternatively damaged parts of mitochondria that undergo fission from the rest of the mitochondria, are transported retrogradely for either lysosomal degradation, or alternatively fusion with healthy mitochondria, in or near the soma (*Hollenbeck, 1993*; *Miller and Sheetz, 2004*; *Maday et al., 2012*; *Maday and Holzbaur, 2014*; *Cheng et al., 2015*; *Zheng et al., 2019*; *Evans and Holzbaur, 2020*). Consistent with these studies, there is evidence that axonal autophagy initiates preferentially at the distal tips or presynaptic sites of axons and terminates the clearance process at the soma (*Yue, 2007*; *Wang et al., 2015*; *Soukup et al., 2016*; *Stavoe et al., 2016*). Indeed, *Pink1* mRNA can be transported to distal regions of neurons and translated locally, providing a constant supply of fresh PINK1 protein for axonal mitochondria, and that this mechanism might enable mitochondria to be at least tagged for degradation in the distal regions of axons (*Harbauer et al., 2022*). However, other studies have demonstrated that axonal mitochondria in mid- or distal-axons tend to be older and more susceptible to damage when compared to those residing in the proximal axons or cell body (*Lehmann et al., 2011*; *Ferree et al., 2013*), and that the vulnerable or stressed axonal mitochondria have diminished mitochondrial motility compromising their ability to return to the soma for degradation (*Wang et al., 2011*; *Lovas and Wang, 2013*; *Ashrafi et al., 2014*; *Hsieh et al., 2016*; *Cheng and Sheng, 2021*), implying that not only initiating but also completing mitophagy locally in the axon may be needed to effectively remove arrested old or damaged mitochondria in long axons.

We previously reported that in the optic nerve head of mice, an alternative mechanism exists to degrade axonal mitochondria, whereby outpocketings of axons containing mitochondria are pinched off from the axons and are degraded by the local astrocytes (*Davis et al., 2014*). Although they occurred elsewhere in the central nervous system, including in the cerebral cortex, these sheddings were most numerous in the optic nerve head, which is both a highly specialized anatomical structure where axons are constantly subjected to biomechanical stress and is also the site of axon damage in glaucoma (*Burgoyne et al., 2005*; *Sigal et al., 2005*). More recently, similar sheddings of mitochondria have been observed in both Parkinson's and Alzheimer's disease rodent animal and cell models (*Morales et al., 2020*; *Lampinen et al., 2022*) as well as in cone photoreceptors (*Hutto et al., 2023*).

OPTN is one of the very few genes that have a large effect on glaucoma (*Rezaie et al., 2002*; *Meng et al., 2012*), and the mutations in OPTN that promote glaucoma but not those that promote Amyotrophic Lateral Sclerosis (ALS) act in a gain-of-function manner to promote more or dysregulated mitophagy (*Wong and Holzbaur, 2014a*; *Sirohi et al., 2015*; *Shim et al., 2016*). Since mutations in mitophagy machinery result in damage to axons in the optic nerve head (*Minegishi et al., 2016*), and this is the location with the highest amounts of transcellular mitochondrial degradation (*Davis et al., 2014*), the current study sought to determine whether mutations in OPTN associated with glaucoma might lead to alterations in the non-cell-autonomous degradation of mitochondria. Since much of

what has been learned about OPTN in mitophagy derives from live-imaging studies of mitochondria and mitophagic machinery in cultured cells (*Lazarou et al., 2015*; *Moore and Holzbaur, 2016*; *Shim et al., 2016*; *Evans and Holzbaur, 2020*), we took a live-imaging approach where the anatomical relationship between axons and their cellular neighbors is maintained, as these cell–cell interactions are likely necessary to reproduce the relevant cell biology. These studies were carried out in young *Xenopus laevis* tadpoles, as this vertebrate model system has two properties optimal for the proposed studies: an optic nerve that can be readily live-imaged and a highly efficient transgenesis suitable for the interrogation of gene variants.

## Results

### Live imaging of mitochondria in RGC axons shows that under basal conditions approximately half of axonal mitochondria are stopped

To live-image RGC mitochondria within axons, we first intravitreally injected Mitotracker Deep Red into *Nieuwkoop and Faber, 1994* stage 48 (10 days post-fertilization) *X. laevis* tadpoles, animals whose RGC axons were readily visible due to expression of an RGC-specific *Tg(Isl2b:GFP)* transgene (*Figure 1A*). After allowing the dye to spread down the axons, the optic nerves of anesthetized and immobilized animals were imaged using a spinning disc confocal microscope (*Figure 1B*, and *Figure 1—video 1*, top) at 1 Hz for 1 min, to measure axonal mitochondria movement through kymographs (*Figure 1C, D*). Since the morphology and movement of mitochondria labeled by Mitotracker were similar to those labeled by an RGC-specific *Tg(Tom20-mCherry)* transgene (*Figure 1—video 1*, bottom), intravitreal Mitotracker injection was deemed suitable for labeling RGC mitochondria in the optic nerve. To determine the stability of Mitotracker labeling over time, optic nerves were imaged 3.5 or 18 hr after intravitreal Mitotracker injection. Since the distribution (*Figure 1—figure supplement 1A*) and velocity (*Figure 1—figure supplement 1B*) of stationary, anterogradely moving, and retrogradely moving mitochondria were similar at both time points, all subsequent imaging experiments were carried out 15–18 hr after Mitotracker intravitreal injection. The kymographs of axonal RGC mitochondria were used to determine the fraction of mitochondria that were stationary, defined here as moving less than 0.1 μm/s, as opposed to moving anterogradely and retrogradely (*Figure 1C, D*). In an analysis of over 500 axonal mitochondria derived from 6 animals, we found that stationary mitochondria account for nearly half of the total axonal mitochondria, whereas both anterogradely and retrogradely transported mitochondria each contribute a quarter of the total (*Figure 1E*), consistent with other studies of axonal mitochondria (*Zheng et al., 2019*; *Fellows et al., 2020*; *Suh et al., 2021*). Among the motile mitochondria, the average speed was 0.63 and 0.78 μm/s for anterogradely and retrogradely moving mitochondria, respectively (*Figure 1F*), which is within the range of the values previously measured by others (*Morris and Hollenbeck, 1993*; *Ligon and Steward, 2000*; *De Vos et al., 2003*; *Miller and Sheetz, 2004*; *Jiménez-Mateos et al., 2006*; *Pilling et al., 2006*; *Mironov, 2007*; *Misgeld et al., 2007*; *Kang et al., 2008*; *Wang and Schwarz, 2009*; *MacAskill and Kittler, 2010*; *Chang et al., 2011*; *Chen et al., 2016*; *Niescier et al., 2016*). Thus, the *X. laevis* tadpole optic nerve was deemed a suitable model to study RGC axonal mitochondria.

Next, to test whether the Mitotracker intravitreal injection itself might affect mitochondria movement within the optic nerve, we crossed *Tg(Isl2b:GFP)* and *Tg(Isl2b:Tom20-mCherry)* animals and obtained doubly transgenic animals which express both a mitochondria-targeted mCherry fluorescent reporter as well as a cytoplasmic GFP transgene, both specifically in RGCs. Mitotracker Deep Red was injected into *Tg(Isl2b:GFP)* expressing tadpoles at NF stage 48, into both animals with or without expression of the Isl2b:Tom20-mCherry transgene, and then the optic nerves were live-imaged as described above within 1 day of the intravitreal Mitotracker injection. The Mitotracker-based quantifications showed no significant difference in mitochondria movement between the two different Mitotracker-injected groups (with and without presence of the mitochondria-targeted transgene), whereas the Tom20-based quantifications showed that the Mitotracker intravitreal injection does have a small but significant effect on slowing retrograde transport and increasing the fraction of stopped mitochondria at the expense of anterograde moving mitochondria (*Figure 1—figure supplement 1F, E*, respectively). Because of this effect, in all subsequent experiments, animals with or without intravitreal injection were compared only to other animals similarly treated.

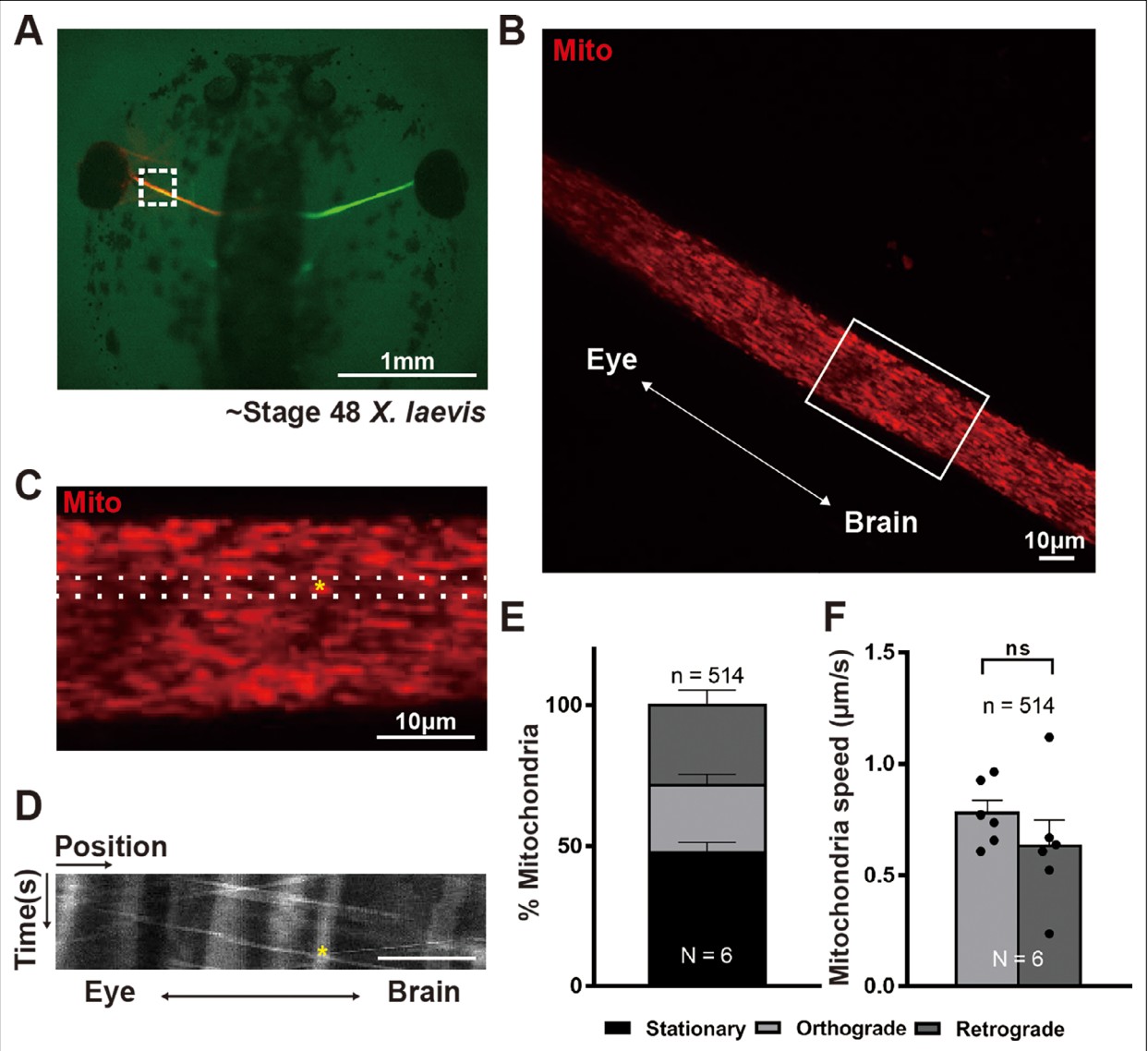

**Figure 1.** Live-imaging mitochondria movement within *X. laevis* tadpole optic nerve shows that in retinal ganglion cell (RGC) axons approximately half of mitochondria are stopped and there is balanced anterograde and retrograde movement. (**A**) Intravitreal injection of Mitotracker Deep Red can be used to label RGC mitochondria within axons; here in *Tg(Isl2b:GFP)* transgenic tadpoles where the RGC axons are also labeled by a cytoplasmic GFP expressed by the *Isl2b* RGC-specific promoter. The white dotted box illustrates the approximate location of spinning-disc confocal live imaging (1 min, 1 Hz). Representative single frame of a Mitotracker Deep Red labeled optic nerve full region imaged in (**B**); boxed area rotated and enlarged in (**C**). (**D**) Representative kymograph displaying the position (*x*-axis) of mitochondria over time (*y*-axis, 60 s); derived from the region between the dotted lines in C for all 60 frames. The yellow star indicates the position of the same mitochondrion in both kymograph (**D**) and representative image (**C**). Scale bar, 10 µm. Quantification of mitochondrial movements, showing that (**E**) about half of RGC axonal mitochondria are stationary and there are comparable numbers of mitochondria moving anterogradely and retrogradely, and (**F**) the average speed of anterograde and retrograde mitochondria movement is similar. Mean ± SEM; *n* = 514 mitochondria from 6 animals. Statistical analysis in (**F**) was performed by unpaired, two-tailed Student's *t*-test. Not significant (ns).

The online version of this article includes the following video and figure supplement(s) for figure 1:

**Figure supplement 1.** Intravitreal Mitotracker injection results in optic nerve mitochondria labeling that is stable within 1 day of injection, labels transgenically labeled retinal ganglion cell (RGC) axonal mitochondria, and has a small but significant effect on the movement behavior of axonal mitochondria.

**Figure 1—video 1.** Intravitreal injection of Mitotracker and retinal ganglion cell (RGC)-expressed Tom20-mCherry transgene similarly labels axonal mitochondria.

https://elifesciences.org/articles/103844/figures#fig1video1

# Doxycycline-induced expression in RGCs of disease-associated OPTN mutants leads to changes in the movement of OPTN and LC3b and, in the case of OPTN E50K, also the movement of mitochondria as well as the co-localization of OPTN and mitochondria

Mutations in OPTN can lead to impaired or aberrant mitophagy and the accumulation of defective mitochondria (*Wong and Holzbaur, 2014a*; *Lazarou et al., 2015*; *Shim et al., 2016*; *Evans and Holzbaur, 2020*). To test whether OPTN mutations might affect the behavior of RGC mitochondria and components of the mitophagic machinery, OPTN and LC3b, specifically within axons, we generated transgenic *X. laevis* in which fluorescently tagged LC3b and variants of OPTN could be expressed selectively in RGCs by induction with doxycycline (*Figure 2A, B*). In these large transgene constructs, regulatory regions from the zebrafish Isl2b gene (*Pittman et al., 2008*) which drive RGC-specific expression in the *X. laevis* retina (*Watson et al., 2012*), were here used to drive a version of the reverse tetracycline trans-activator, rtTA2, that is highly efficient in *X. laevis* (*Das and Brown, 2004*; *Mills et al., 2015*) in plasmids that also contain a bidirectional tetracycline operator that expresses EGFP-LC3b in one direction and in the other mCherry-tagged versions of the following human OPTN variants: Wt, glaucoma-associated mutations E50K, M98K, and H486R, ALS-associated mutation E478G, and two synthetic mutations, F178A and D474N, previously shown to be defective in the OPTN interaction with LC3b and ubiquitinated mitochondria, respectively (*Wild et al., 2011*; *Korac et al., 2013*; *Wong and Holzbaur, 2014a*; *Sirohi et al., 2015*; *Shim et al., 2016*; *Li et al., 2018*; *Liu et al., 2018*; *Chernyshova et al., 2019*; *Padman et al., 2019*). The optic nerves of 4–6 transgenic tadpoles per transgene were imaged using the spinning-disc confocal first as a 1-min single plane t-series acquired at 1 Hz, and then as a z-scan imaging the full thickness of the optic nerve at 1 μm steps. In the case of animals expressing E50K and Wt OPTNs, analyses were done with and without previous labeling of RGC mitochondria through intravitreal injection of Mitotracker (*Figure 2C, D*); in the case of all other OPTN variants, analyses of OPTN and LC3b were carried only without Mitotracker labeling. Kymograph analyses of the t-series were used to quantify the percentage of each movement class (stationary, anterograde, and retrograde) and determine the average speed of anterograde and retrograde movement. For the live-imaging and kymograph analyses of OPTN and LC3b, in the absence of Mitotracker intravitreal injection, analyses were carried out in both primary transgenic F0 animals (*Figure 2E, E'*, *Figure 2—figure supplement 1C, D*) as well as in their F1 progeny (*Figure 2—figure supplement 1A, B, E, F*), to minimize the possibilities of transgene copy number or integration position effects confounding the interpretation. Such analyses in F0s are possible in *X. laevis* because the restriction enzyme-mediated integration transgenesis method used to generate the animals (*Kroll and Amaya, 1996*) involves transgene integration prior to the first cell division, and thus provides fully transgenic rather than mosaic animals. The results from F0 transgenic founders and F1 progeny were similar and showed that relative to Wt OPTN, all OPTN mutants with the exception of the LC3b-binding-defective F178A increased the fraction of stopped OPTN or LC3b in either F0 or F1 experiments. However, it was only the glaucoma-associated E50K and M98K mutations that increased stopped OPTN in both F0 and F1 analyses, and only the E50K mutation that increased both OPTN and LC3b in both F0 and F1 analyses. Analyses of anterograde and retrograde velocity for OPTN and LC3b in both F0s (*Figure 2—figure supplement 1C, D*) and F1 animals (*Figure 2—figure supplement 1E, F*) were generally consistent with one another. Overall, there were no consistent significant changes in velocity in both F0 and F1 analyses, suggesting the increases in stopped OPTN and LC3b were specific effects and not secondary to making the RGCs or their axons unhealthy. To determine whether the observed increases in stopped OPTN and LC3b might represent an increase in axonal mitophagy, the studies of OPTN and LC3b movement were repeated, but now 15–18 hr after intravitreal injection of Mitotracker (*Figure 2F*); these studies were carried just in F1 animals expressing Wt or E50K OPTN, as well as in animals carrying no transgenes at all. Compared to the non-transgenic control animals (only Mitotracker-injected), those with expression of Wt OPTN had no alteration in the fraction of stopped mitochondria, similar to what had been previously observed after expression of the Tom20-mCherry transgene (*Figure 1—figure supplement 1E*), though they did have a small effect on the velocity of retrogradely transported mitochondria (*Figure 2—figure supplement 1G*); this effect on retrograde velocity was similar after both Wt and E50K OPTN expression. In contrast, expression of E50K OPTN uniquely resulted in a significant increase in the fraction of stopped mitochondria, above the already high baseline amount (*Figure 2F*), without affecting retrograde transport

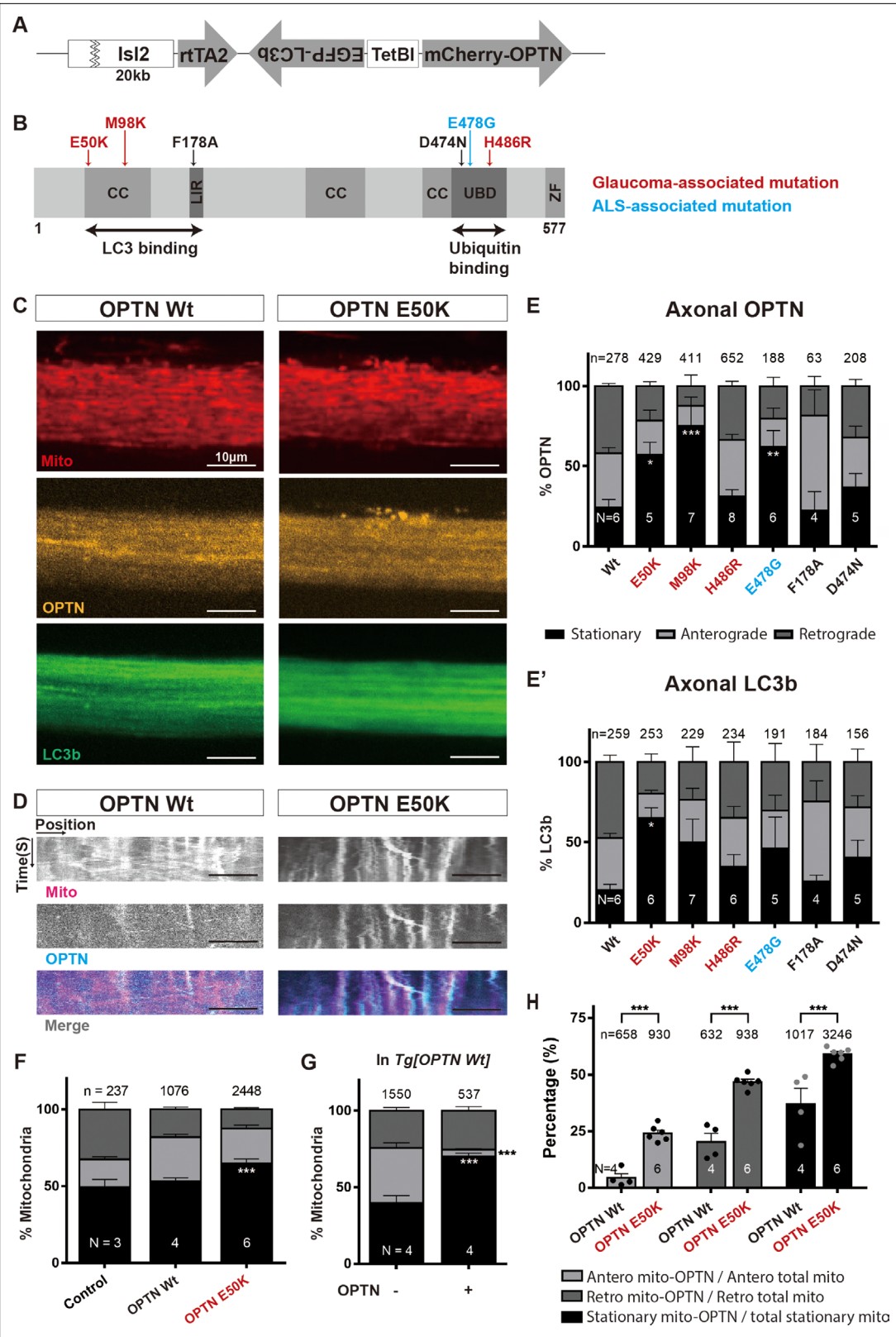

**Figure 2.** OPTN mutants conditionally expressed in retinal ganglion cells (RGCs) increase the fraction of stopped axonal mitochondria, OPTN and LC3b, and the fraction of mitochondria co-localizing with OPTN. (**A**) Illustration of the transgenic construct: Isl2b promoter driving rtTA2 linked to EGFP-LC3b and mCherry-OPTN driven in opposite strands by a bidirectional tetracycline operator, TetBI. The fusion constructs are thus expressed only in RGCs and only after doxycycline induction. (**B**) Schematic of OPTN functional domains showing the position of the point mutations examined.

*Figure 2 continued on next page*

*Figure 2 continued*

Three glaucoma-associated mutations, E50K, M98K, and H486R; an ALS-associated mutation, E478G; two synthetic OPTN mutations, F178A and D474N, disrupting LC3b and ubiquitin binding, respectively. CC: coiled coil domain, LIR: LC3 interacting region, UBD: ubiquitin-binding domain, ZF: zinc finger. Numbers indicate the amino acid position. Representative confocal images (**C**) and corresponding kymographs (**D**) of axonal mitochondria, OPTN, and LC3b in Mitotracker-injected *Tg(Isl2b:mCherry-OPTN(Wt or E50K)_ EGFP-LC3b)* animals 3 days after induction of transgene expression. Scale bar, 10 μm. Quantification of OPTN (**E**) and LC3b (**E'**) movements in Wt OPTN and various OPTN mutants (all independent F0 animals). Bar graphs show the percentage of each movement (stationary, anterograde, and retrograde) of OPTN and LC3b in Wt OPTN and OPTN mutants. Mean ± SEM; *n* = 63–652 OPTN and *n* = 156–259 LC3b objects measured in 4–8 animals. (**F**) Quantification of mitochondrial movements after Mitotracker injection in control (non-transgenic), and transgenic animals expressing in RGCs either Wt or E50K OPTN (analyzed in F1 animals). Expression of E50K OPTN but not Wt OPTN results in a significant increase in stalled mitochondria compared to the control. Mean ± SEM; *n* = 237–2448 mitochondria from 3 to 6 animals. Quantification of mito-OPTN co-localization in Mitotracker-injected Wt OPTN (**G**, **H**) and E50K OPTN (**H**) animals. (**G**) Percentage of each movement (stationary, anterograde, and retrograde) of mitochondria co-localizing (mito-OPTN) or not co-localizing (mito-ONLY) with OPTN in the animals expressing Wt OPTN. Mean ± SEM; *n* = 1550 mito-ONLY, *n* = 537 mito-OPTN co-localizations from 4 Wt OPTN animals. (**H**) Expression of E50K OPTN results in increased fraction of mito-OPTN (mitochondria-OPTN co-localization) relative to Wt OPTN, both in the moving and stationary pools. Mean ± SEM; *n* = 658–930 total anterograde mitochondria, *n* = 632–938 total retrograde mitochondria, *n* = 1017–3246 total stationary mitochondria from 4 to 6 animals. Statistical analysis in (**E**–**H**) was performed by two-way ANOVA following Tukey's post hoc test for multiple comparisons. *p < 0.05, **p < 0.01, ***p < 0.001.

The online version of this article includes the following figure supplement(s) for figure 2:

**Figure supplement 1.** Movement of OPTN and LC3b in F1 animals (not injected intravitreally with Mitotracker) matches those observed in F0 animals, including showing no large velocity changes in either anterograde or retrograde movement.

any further (*Figure 2—figure supplement 1G*). Of note, as was the case for OPTN and LC3b in the absence of intravitreal Mitotracker injection, expression of neither Wt nor E50K OPTN affected the relative balance of retrograde versus anterograde movement of mitochondria, OPTN, or LC3b but rather only affected the fraction of the stopped populations. Consistent with the previous results showing that intravitreal injection of Mitotracker had a small but significant effect on mitochondria movement (*Figure 1—figure supplement 1E, F*), here too we found that the intravitreal injection of Mitotracker affected the behavior of axonal OPTN, including an increase in the fraction of stopped OPTN; however, so did animals receiving an injection of just the biological salts solvent, showing that it is the eye injection procedure rather than the Mitotracker itself that modestly perturbs the system (*Figure 2—figure supplement 1H*).

Next, to determine whether the increases in stopped mitochondria and stopped OPTN might be linked, the degree of mitochondria and OPTN co-localization was examined. In the nerves labeled by Mitotracker, visual inspection of the raw images (*Figure 2C*) and the derived kymographs (*Figure 2D*) was suggestive that OPTN and the Mitotracker-labeled mitochondria might co-localize, particularly in the stopped populations, and more so in the animals expressing E50K OPTN. To examine the degree of mitochondria and OPTN co-localization quantitatively, we re-analyzed the previous kymographs of mitochondria and OPTN and separately quantified mitochondria that co-localized, with OPTN (mito-OPTN) versus mitochondria not associated with OPTN (mito-ONLY) (*Figure 2G*, *Figure 2—figure supplement 1I*). Strikingly, while mitochondria not co-localizing with OPTN had relatively balanced populations of stopped, and anterogradely and retrogradely moving, the majority of mitochondria co-localizing with OPTN were stopped, with the remainder being largely engaged in retrograde transport; the same was true in animals expressing Wt OPTN (*Figure 2G*) and E50K OPTN (*Figure 2—figure supplement 1I*). The glaucoma-associated E50K OPTN also significantly increased the fraction of mitochondria co-localizing with OPTN, and this increased co-localization was observed not only in the stationary but also the two moving populations (*Figure 2H*). In summary, expression of OPTN versions carrying glaucoma and ALS-associated mutations, as well as a synthetic mutation in OPTN that perturbs the recognition of ubiquitinated mitochondria, but not a synthetic mutation that perturbs the association of OPTN with LC3b, all had or trended to increases in stopped axonal OPTN and LC3b in RGCs, and the two that had the largest and most consistent effects were two glaucoma-associated mutations, E50K and M98K, with E50K having an increased co-localization with mitochondria and leading to a greater fraction of stopped mitochondria within the optic nerve.

Since OPTN associates with damaged mitochondria when mediating mitophagy, we asked whether some of the stopped mitochondria co-localizing with OPTN measured in our kymographs might be mitochondria undergoing mitophagy. If so, inhibiting autophagosome-lysosome fusion with chloroquine would be expected to affect the number of these stopped mitochondria. Surprisingly,

as measured by the Tom20-mCherry mitochondria reporter, neither the total number (*Figure 3—figure supplement 1A*) nor the proportions of mitochondria stopped or moving in either direction (*Figure 3—figure supplement 1B*) were affected by 24 hr of chloroquine treatment. These negative results lead us to confirm the efficacy of chloroquine in our system by using transgenic animals expressing the autophagosome/autophagolysosome reporter LC3b-EGFP-mCherry, expressed in astrocytes through a promoter from the fatty acid binding protein 7 gene (*fabp7*, also known as basic lipid binding protein or *blbp*) (*Owada et al., 1996*; *Sharifi et al., 2011*), which drives expression in frog optic nerve astrocytes (*Mills et al., 2015*). While discrete green or red puncta were rare, chloroquine did result in the expected increase in green puncta that report autophagosomes that have not yet fused with lysosomes (*Figure 3—figure supplement 1C*). Within some of those autophagosomes, there was Mitotracker signal (*Figure 3—figure supplement 1D*), suggesting that at least some axonal mitochondria might be degraded by astrocyte autophagolysosomes. To quantify the extent of RGC axonal mitochondria found in the optic nerve outside of the axons, the same chloroquine treated animals were analyzed as z-scans of the full thickness of optic nerves acquired at 1 μm steps, followed by measurements using 3D imaging software Imaris of the Tom20-mCherry signal inside or out of a mask based on a membrane GFP transgene expressed by the same axons (*Figure 3—figure supplement 1E, F*). Such analyses showed that 21.4% of the Tom20-mCherry signal is outside of the axons, much of it near the surface of the optic nerve, but that amount is not significantly altered by 24 hr of chloroquine treatment (*Figure 3—figure supplement 1G*). Note that this measure of axonal mitochondria outside axons likely relates to only the stopped mitochondria, as the z-scan imaging used frame-averaging and thus moving objects were blurred and likely underrepresented. Thus, in our system, the stopped mitochondria co-localizing with OPTN appear not to be degraded by conventional mitophagy.

Since in many of the animals some OPTN signal appeared not to be co-localized with the LC3b signal but rather to also be on the surface of the optic nerve (e.g., middle top of the OPTN E50K nerve shown in *Figure 2C*), similar to what was observed with the axon expressed Tom20-mCherry transgene, we then asked whether significant amounts of OPTN might also be found outside of axons, and if so, whether the OPTN mutations affected how much. Because LC3b localizes not only to autophagosomal membranes, but also is in the cytoplasm/axoplasm (*Xie and Klionsky, 2007*; *Mizushima and Komatsu, 2011*; *Fu et al., 2014*; *Wong and Holzbaur, 2014b*; *Tammineni et al., 2017*; *Stavoe et al., 2019*; *Evans and Holzbaur, 2020*; *Boecker et al., 2021*; *Kuijpers et al., 2021*), we used the EGFP-LC3b signal to create a mask to demark the RGC axons, so as to then be able to separately measure the OPTN (mCherry) or mitochondria (far-red) signals that co-localized or not with the axons. First, such analyses of OPTN outside axons were carried out for the same animals above analyzed for OPTN and LC3b movement metrics, but that had not been subjected to the intravitreal injections of Mitotracker, animals in which the expression of mCherry-hOPTNs (Wt, E50K, M98K, F178A, D474N, E478G, and H486R) and EGFP-LC3b was induced in RGCs through bath application of doxycycline. Measurements carried out in the 3D reconstructions showed significant increases in the fraction of OPTN outside of the RGC axons in the optic nerves of glaucoma-associated mutations, especially E50K (12.6%), when compared to Wt OPTN (0.5%), ALS-associated mutation E478G (0.2%), and the synthetic mutations F178A (3.3%) and D474N (0.1%) (*Figure 3A, B*). Then, to determine whether such extra-axonal OPTN might be associated with mitochondria, the same procedure was applied to the z-scan images of animals where Wt OPTN and E50K OPTN were similarly induced by doxycycline, but that also a day prior had received an intravitreal injection of Mitotracker (*Figure 3C*). Imaris-based 3D reconstruction and volume quantification revealed that expression of OPTN carrying the glaucoma-associated E50K mutation results in a significant increase in both the fraction of RGC axonal mitochondria outside of the LC3b-labeled RGC axons (13.4%) but also the fraction of all stopped RGC axonal mitochondria co-localizing with OPTN also outside of the LC3b-labeled RGC axons (17.9%), as compared to the much lower amounts found after expression of Wt OPTN (2.7% for mitochondria and 0.5% for mito-OPTN) (*Figure 3D, E*). Note that the amount of mitochondrial signal measured outside of axons based on Mitotracker labeling is less than that estimated to be outside based on the Tom20mCherry transgene. In summary, expression of E50K OPTN, but not OPTN that lacks the glaucoma-associated mutation, not only increases the amount of stopped mitochondria, OPTN and LC3b, and the degree of co-localization between mitochondria and OPTN within axons but also increases the amounts of mitochondria and

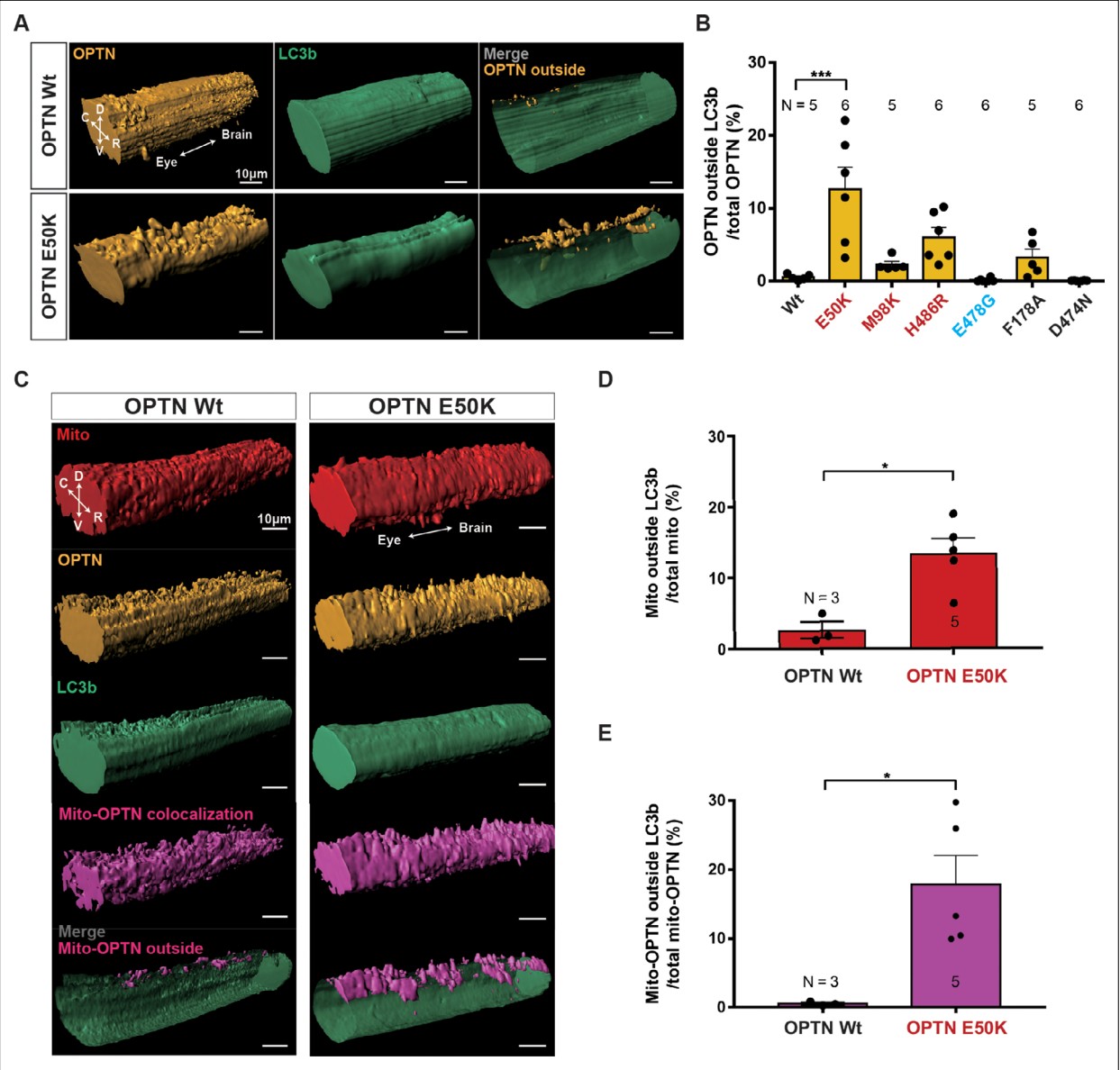

**Figure 3.** 3D reconstruction of axonal mitochondria, OPTN and LC3b in optic nerves shows that glaucoma-associated OPTN mutations increase the amounts of mitochondria, OPTN, and mito-OPTN outside of the LC3b-labeled axons. (**A**) Reconstruction of optic nerves displaying OPTN, LC3b, and OPTN outside LC3b-labeled retinal ganglion cell (RGC) axons (merged images on the right) in the optic nerves of animals expressing Wt and E50K OPTN. D: dorsal, V: ventral, C: caudal, R: rostral. Scale bar, 10 μm. (**B**) Percentage of OPTN outside LC3b-labeled RGC axons is significantly higher or trends so in the glaucoma-associated OPTN mutations. Mean ± SEM; $n$ = 5–6 animals. (**C–E**) 3D reconstruction of axonal mitochondria, OPTN, and LC3b within the segments of intraocular optic nerves in Mitotracker-injected *Tg(Isl2b:mCherry-OPTN(Wt or E50K)_ EGFP-LC3b) X. laevis.* (**C**) Reconstruction displaying mitochondria, OPTN, LC3b, mito-OPTN co-localization, and mito-OPTN outside LC3b-labeled RGC axons (merged images on the bottom) in the optic nerves of animals expressing Wt and E50K OPTN. D: dorsal, V: ventral, C: caudal, R: rostral. Scale bar, 10 μm. Percentage of mitochondria (**D**) and mito-OPTN (**E**) outside of LC3b-labeled RGC axons after expression of Wt and E50K OPTN. Expression of E50K OPTN results in a significant increase in mitochondria and mito-OPTN co-localizations outside of LC3b-labeled RGC axons. Mean ± SEM; $n$ = 3–5 animals. Statistical analysis in (**B**) was performed by one-way ANOVA following Tukey's post hoc test for multiple comparisons, and (**D, E**) was performed by unpaired, two-tailed Student's *t*-test. *$p < 0.05$, ***$p < 0.001$.

The online version of this article includes the following video and figure supplement(s) for figure 3:

**Figure supplement 1.** Inhibiting autophagolysosome formation through chloroquine minimally affects retinal ganglion cell (RGC) axonal mitochondria degradation.

**Figure 3—video 1.** E50K OPTN expression increases the amount of MitoOPTN outside of axons.

https://elifesciences.org/articles/103844/figures#fig3video1

the mitophagy receptor OPTN being found outside of the axons, much of which also co-localize outside of the axons.

## Sparse labeling of axons shows even larger amounts of axonal mitochondria and OPTN are outside of axons and that only in the most extreme cases are they observed on the ON surface

In the experiments just described, much LC3b-negative-OPTN-positive mitochondria signal was found on what appears to be the surface of the optic nerve, which was most obvious after expression of E50K OPTN (see *Figure 3A, C*, and *Figure 3—video 1*). To determine whether additional extra-axonal OPTN and mitochondria might be found within the optic nerve parenchyma but obscured by the EGFP-LC3b mask due to the high axon density and low imaging resolution, we turned to a sparse labeling approach. Small groups of retinal cells, estimated to be 10–50 in number, were surgically transplanted from E50K OPTN transgenic progeny into progeny of a line expressing *Tg(Fab-p7:mTagBFP2-Ras)*, in which astrocyte membranes are labeled by a membrane-targeted BFP reporter. Transplants were done around NF stage 24–32, when all cells in the retina are progenitors and thus prior to RGC differentiation. A week later in such animals and 3 days after doxycycline induction, small groups of axons were readily visible in these optic nerves based on the EGFP-LC3b signal. 3D reconstruction and volume measurements of the astrocyte membrane signal derived from the host and the OPTN and LC3b signals derived from E50K OPTN expressing donor cells (*Figure 4A, B*, and *Figure 4—video 1*) showed that the fraction of OPTN signal on the surface of the optic nerve was 17.1% of the total signal, in relatively good agreement with the data previously obtained in the whole nerves (see *Figure 3B*). However, this represented only 23.9% of the total OPTN outside of the LC3b-labeled axons, the rest being within the optic nerve parenchyma. Furthermore, the nerves with the largest amounts of OPTN on the surface of the optic nerve were those with the greatest amount of extra-axonal OPTN (*Figure 4B*). These sparse labeling data thus suggested that our previous measures of OPTN and mitochondria outside of axons quantified in *Figure 4* based on the whole-nerve imaging after expression of all OPTN variants likely grossly underreported the actual amounts of OPTN and mitochondria outside of the axons. Thus, for animals expressing either Wt or E50K OPTN, we repeated the same t-series and z-scan imaging and following kymograph analyses and Imaris 3D reconstructions in the context of the sparse labeled axons, but also after dye-labeling the mitochondria prior to transplantation. That is, 1 hr after injection of Mitotracker into the eye anlage, retina progenitor cells from Wt or E50K OPTN transgenic progeny were transplanted into the eyes of non-transgenic hosts (*Figure 4C*, *Figure 4—figure supplement 1A, B*). Consistent with our previous data, more stalled mitochondria, OPTN, and LC3b puncta were observed in the axons expressing E50K OPTN compared to those expressing Wt OPTN, with relatively balanced populations of antero-grade and retrograde movement in both Wt and E50K OPTN (analyzed per axon in *Figure 4—figure supplement 1C* and analyzed per animal in *Figure 4—figure supplement 1C'*). Anterograde and retrograde velocities for mitochondria, OPTN, and LC3b in the sparsely labeled Wt and E50K OPTN axons also were similar to those measured in the whole nerves, and once again showed no significant differences in velocities after expression of E50K OPTN (*Figure 4—figure supplement 1D*). Of note, the diameter of axons based on the LC3b signal (*Figure 3—figure supplement 1E*) and the inten-sity of the LC3b signal (*Figure 3—figure supplement 1F*) were comparable after expression of Wt or E50K OPTN, validating the use of cytoplasmic LC3b-GFP as a suitable axon mask in both these sparse axon experiments and the previous whole nerve experiments. Consistent with the other sparse labeling results above, the amounts of mitochondria and OPTN outside of axons were higher than those previously measured. In the case of E50K OPTN, 35.8% of mitochondria and 21.8% of OPTN were determined to be outside of the axons in the sparse labeling experiments (as compared to 13.4% and 17.9%, respectively, measured in the whole nerves). The difference between whole nerve and sparsely labeled axons was far larger in the case of expression of Wt OPTN, where previously approximately 2.7% and 0.5% had been found for mitochondria and mito-OPTN, respectively (see *Figure 3D, E*), but now 15.2% of mitochondria and 5.7% of OPTN were shown to be outside of the RGC axons (*Figure 4D, E*); this order of magnitude difference in results is presumably due to nearly all the extra-axonal Mitotracker-labeled mitochondria and OPTN are found within the nerve parenchyma, not having reached high enough levels to be measurable on the surface of the optic nerve. Collec-tively, these data show that even in the case of expression of Wt OPTN, which we showed above does

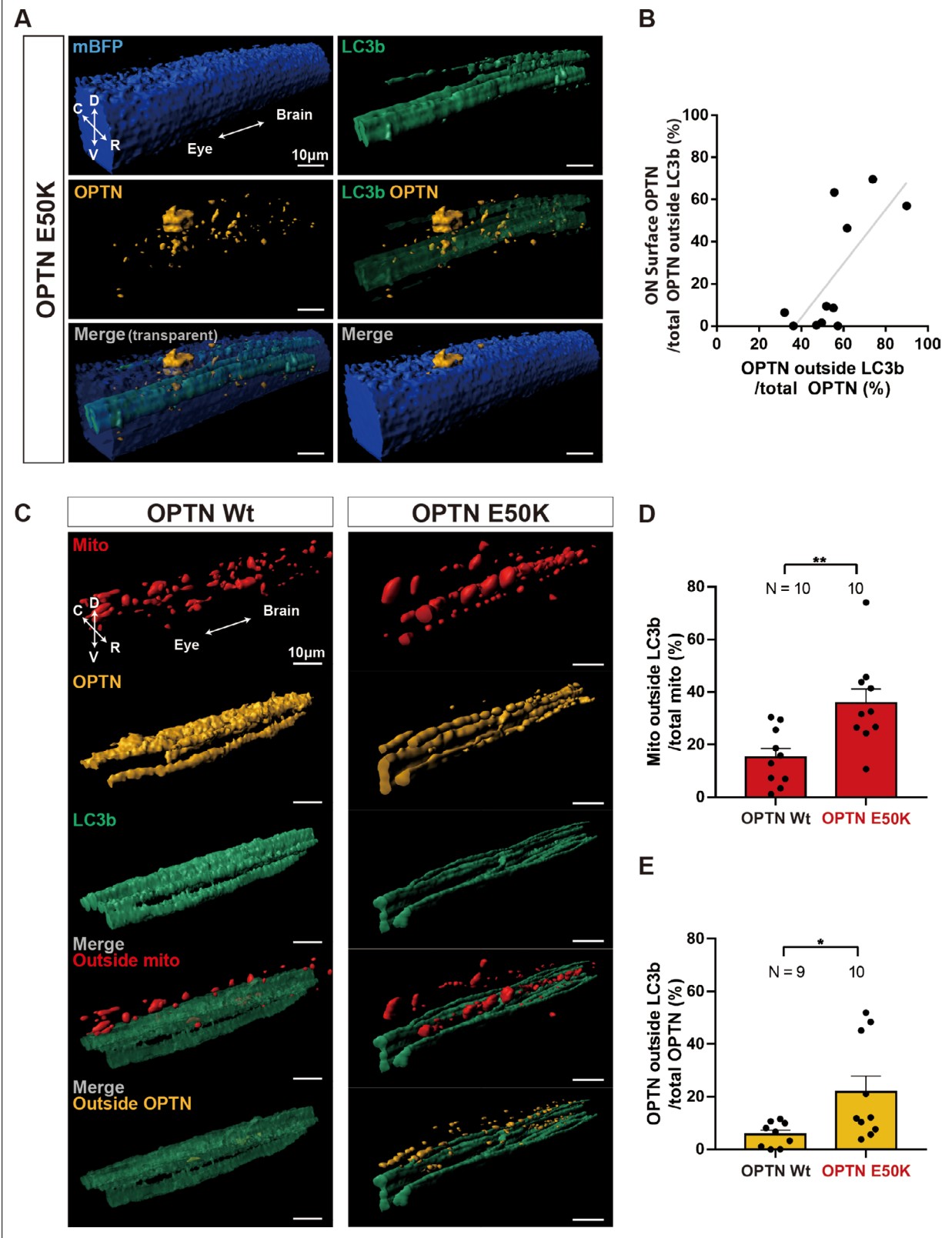

**Figure 4.** Retinal ganglion cell (RGC) axonal mitochondria and OPTN found outside of the LC3b-labeled axons are both within the optic nerve parenchyma and on the surface of the optic nerve and are increased by expression of E50K OPTN. (**A**, **B**) E50K OPTN from sparsely labeled axons found outside of LC3b-labeled axons is found both within the optic nerve parenchyma and at the optic nerve surface. (**A**) Reconstruction showing astrocyte membranes of the host, from *Tg(Fabp7:mTagBFP2-Ras)* here abbreviated as mBFP, and E50K OPTN and LC3b (merged images on the bottom) from the

*Figure 4 continued on next page*

*Figure 4 continued*

axons of the donor cells. D: dorsal, V: ventral, C: caudal, R: rostral. Scale bar, 10 μm. (**B**) Percentage of OPTN on the optic nerve surface (*y*-axis) versus percentage of total OPTN outside of the LC3b-labeled axons (*x*-axis). *n* = 11 animals. Trend line with 0.522 *R*-squared value. (**C, E**) Sparse labeling of axons reveals that extensive amounts of axonal mitochondria and OPTN are outside of the axons. (**C**) 3D reconstructions of sparsely labeled axons showing mitochondria and either Wt or E50K OPTN reveal much of both are outside the LC3b-labeled RGC axons (merged images on the bottom). D: dorsal, V: ventral, C: caudal, R: rostral. Scale bar, 10 μm. Percentage of mitochondria (**D**) and OPTN (**E**) outside the LC3b-labeled RGC axons in animals expressing Wt and E50K OPTN, respectively. Mean ± SEM; *n* = 9–10 animals. Statistical analysis in (**D**) and (**E**) was performed by unpaired, two-tailed Student's *t*-test. *p < 0.05, **p < 0.01.

The online version of this article includes the following video and figure supplement(s) for figure 4:

**Figure supplement 1.** Movements of axonal mitochondria, OPTN, and LC3b within the sparsely labeled Mitotracker-labeled *Tg(Isl2b:mCherry-OPTN(Wt or E50K)_ EGFP-LC3b)* axons of retinal ganglion cells (RGCs) transplanted into non-transgenic *X. laevis*.

**Figure 4—video 1.** Sparse axon labeling reveals additional OPTN outside of axons.

https://elifesciences.org/articles/103844/figures#fig4video1

not alter the movement behavior of mitochondria within axons, there already is a substantial amount of both mitochondria and OPTN outside of the axons from which they derive, and that expression of E50K OPTN, in addition to resulting in the stalling of large number of mitochondria, OPTN, and LC3b within axons and co-localization of mitochondria and OPTN also significantly increase the amount of both OPTN and mitochondria that are found outside of the axons and further result in over an order of magnitude increase in how much mitochondria signal is found near the surface of the optic nerve.

## Shed axonal mitochondria are degraded by astrocytes

To determine where outside axons the axonal mitochondria were degraded, we carried out correlated light EM analyses on sparsely labeled axons. First, cells from animals expressing a membrane-targeted mCherry transgene in RGCs were transplanted into animals expressing two different transgenes in astrocytes, the same membrane-associated BFP transgene used above to label all astrocyte membranes, and also an Aquaporin-4-GFP (Aqp4-GFP) fusion construct that labels principally the glial limitans at the surface of the optic nerve. One day after labeling the RGC axonal mitochondria through an intravitreal injection of Mitotracker Deep Red, the optic nerves of these animals were live-imaged in their entirety at lower *x*-, *y*-, and *z*-resolution for just the Aqp4-GFP transgene (*Figure 5A*), and then one region imaged at higher resolution in all four channels (*Figure 5B*). The heads of fixed animals, heavy metal processed and embedded in epoxy resin, were then analyzed by micro-CT in order to identify the same region that had been live imaged (*Figure 5C*) and then the full thickness of the optic nerve at that location was analyzed by serial block-face scanning electron microscopy (SBEM). After registering the live imaging and SBEM datasets based on the position of nuclei, some of the Mitotracker signal outside axons that had been live-imaged could be unambiguously identified at the level of EM. The axon-derived material on the surface of the optic nerve, in this case dye-labeled mitochondria and transgene-labeled axonal membranes, was determined to be within the soma of a transgene-labeled astrocyte (*Figure 5D, E*), and morphologically appeared as mitochondria and membranes at various stages of degradation. The mitochondria that live-imaging determined to be stationary within the nerve parenchyma were harder to unambiguously correlate between light and SBEM datasets, but those that could were determined to be within the fine processes of astrocytes (*Figure 5F*). Notably, while axonal debris was most often observed within individual fine processes, and those processes were closely apposed to fine processes of distinct astrocytes (*Figure 5F*). The soma of the astrocytes was located at the optic nerve periphery but had processes that extended deep into the parenchyma (*Figure 5G*), similar to astrocytes that surround axon bundles within larger mammalian optic nerves. Thus, it is possible that the axon-derived debris found within astrocyte soma at the surface of the optic nerve might have originated deep within the optic nerve parenchyma. However, it also may have originated near the astrocyte soma, as dystrophic axons with mitochondria undergoing early steps of mitophagy could also be observed within some axons in close contact with the astrocyte soma (*Figure 5H, I*).

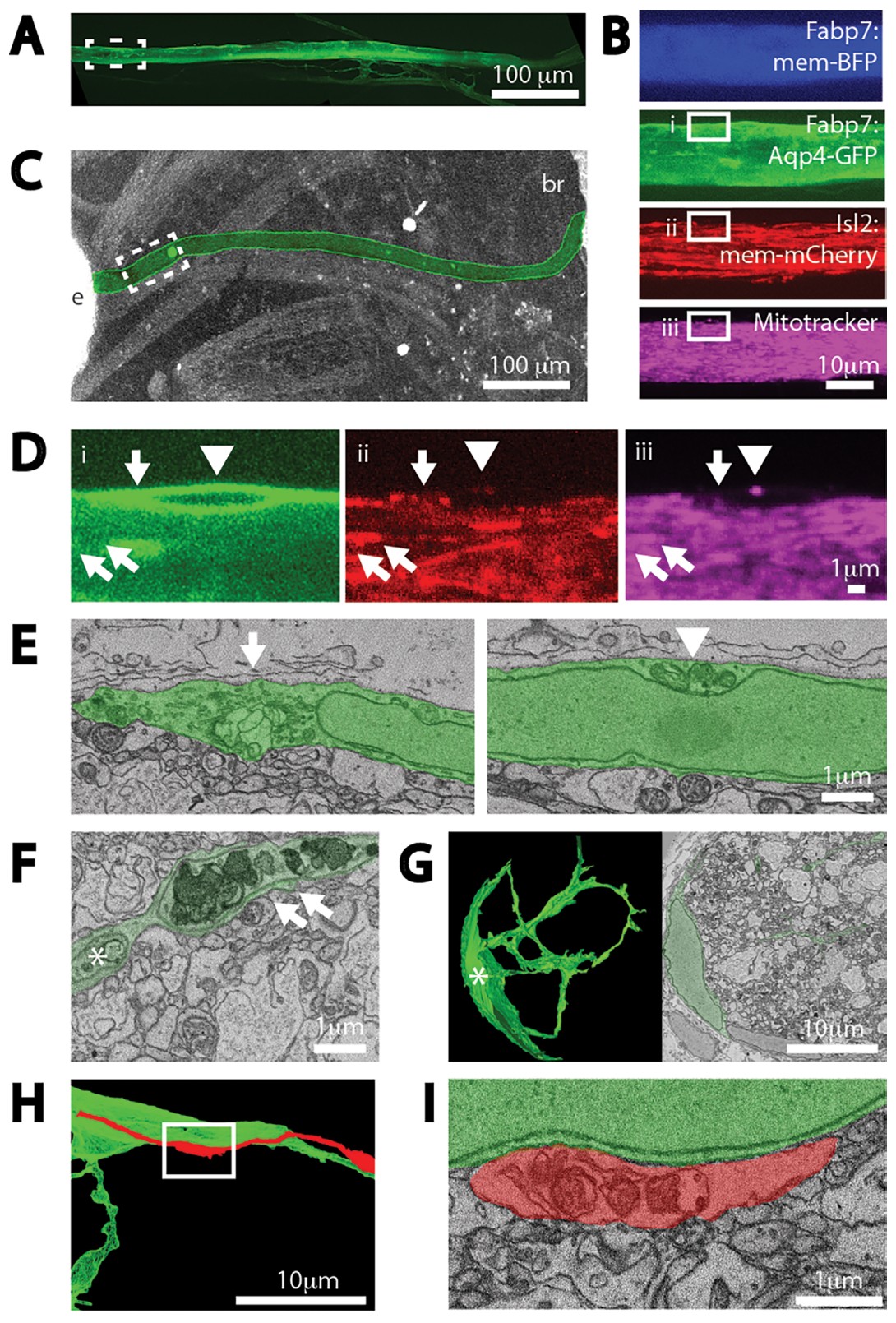

**Figure 5.** Extruded axonal mitochondria are degraded by astrocytes. (**A**) Z-projection of full length of the optic nerve visible through live imaging, labeled by an Aqp4-GFP transgene expressed in astrocytes. (**B**) Z-projections of four channels after higher resolution imaging of a region of the same optic nerve, position shown in stippled boxed region in (**A**); four channels are astrocyte expressed mem-BFP and Aqp4-GFP transgenes, a retinal ganglion cell (RGC) expressed mem-mCherry transgene, together with intravitreally injected Mitotracker. Solid boxes represent fields shown at higher

*Figure 5 continued on next page*

*Figure 5 continued*

resolution in (**D**) and (**E**). (**C**) Micro-CT of the head of the same animal after embedding in epoxy resin, used to pin-point the area live-imaged for subsequent SBEM analyses. Optic nerve is colorized green. Letter insets: e is eye and br is brain. (**D**) Superficial astrocyte soma labeled by astrocyte expressed Aqp4-GFP transgene (i) containing axonal derived membranes (ii) and mitochondria (iii). Arrow, arrowhead, and double arrows represent three discrete axonal mitochondria signal unambiguously identifiable at both light level live imaging and at the level of SBEM, shown in (**E**) and (**F**). (**E**) Two SBEM sections of the same astrocyte soma centered on discrete pockets phagocytosed axonal membranes and mitochondrial material. (**F**) Extra-axonal mitochondria and membranes within fine astrocyte processes. Asterisk represents a process originating from the same astrocyte shown in (**G**). (**G**) Morphology of one of the astrocytes whose processes are near the extra-axonal mitochondria shown in (**F**); on right, reconstruction of about half the astrocyte and on left single plane. (**H**) Reconstruction of an axon near the soma of the same astrocyte shown in (**D**) and (**E**). (**I**) Dystrophic mitochondria within axon appearing to be enwrapped in other axonal membranes within the axon, adjacent to superficial astrocyte soma.

## At least some OPTN and mitochondria leave RGC axons through focal axon dystrophies

To investigate how RGC axonal mitochondria and OPTN leave axons, we performed additional sparse labeling experiments. Live imaging of sparsely labeled axons in animals expressing both a membrane-targeted GFP (lyn-GFP) and a mitochondrial-targeted mCherry (Tom20-mCherry) transgene revealed extensive mitochondria signal outside of axons that was relatively unchanged over 10 min of imaging (*Figure 6A*, and *Figure 6—video 1*). In these nerves, 6.3 ± 0.1% (SD) of the Tom20-mCherry mitochondria signal was outside of the lyn-GFP-labeled axons, which is comparable to the measure of mitochondria outside sparse axons obtained based on Mitotracker labeling. Manual counting of Tom20-mCherry-labeled mitochondria within five discrete axon segments (100–150 µm in length) in two animals and Imaris-based semi-automatic counts within approximately 235 axon segments (75–175 µm in length) in 7 animals provided similar estimates for the frequency of stopped mitochondria, namely 3.8 ± 2.3 (SD) or 6.3 ± 5.0 (SD) per 100 µm of axon length, respectively. Among these sparsely labeled axons, there were also membranous asymmetric focal dystrophies, or protrusions, containing mitochondria similar in shape to what we previously described in the mouse optic nerve head (*Figure 6B*). In the same axons where mitochondria number was counted, a total of 78 of such asymmetric dystrophies were found, thus representing 0.47 per 100 µm length of axon. However, of these protrusions, only 47, or 60.3%, contained mitochondria. Thus, approximately 1 in 20 stopped mitochondria within axons is contained within such protrusions. Of note, the mitochondria that were contained within these axonal protrusions were significantly more spherical than the average axonal mitochondria (*Figure 6C*). Tracking of 18 of these same mitochondria containing protrusions over approximately 8 min of repeated imaging every minute found 13 of them still present at the last time point, showing that these asymmetric mitochondria containing axonal dystrophies persist on average more than 5 min.

In order to determine the localization of OPTN relative to these axonal dystrophies, we performed similar experiments where instead donors expressed LC3b-EGFP and OPTN-mCherry transgenes. Since transplants are challenging to perform and must be done before knowing whether the transplanted cells will express the transgenes, and the intermediates are short-lived, these studies were conducted mainly for axons expressing the glaucoma-associated M98K OPTN mutation, as these animals had the greatest number of stopped OPTN in both F0 and F1 analyses (*Figure 2E*, *Figure 2—figure supplement 1A*). After the repetitive z-scan live imaging of optic nerve sub-volumes, areas with focally increased LC3b and OPTN signal were digitally extracted and their position at different times registered, to measure shape and fluorescence intensity changes over time. Areas of increased LC3b and OPTN were found in four types of structures: symmetric axonal swellings with static LC3b and OPTN, asymmetric axonal dystrophies with static or changing levels of OPTN or LC3b, a small number of asymmetric axonal dystrophies that pinched off from axons during the imaging window, and structures with no connection to axons. An axonal 'swelling' is shown where the mitophagy machinery OPTN and LC3b are focally stopped and enriched together (*Figure 6D, i*). An asymmetric axon dystrophy is shown in what appears to be a 'loading' stage (*Figure 6D, ii* and *Figure 6—video 2*). Here, the OPTN signal increases within the dystrophy simultaneously with a decrease in the OPTN signal immediately below the dystrophy within the associated axon, suggestive of an active loading process (*Figure 6E*). The last example shown is a 'pinching-off' event where the axon dystrophy is physically detached from the axon of origin and appears to amalgamate with other areas of extra-axonal OPTN (*Figure 6D, iii* and *Figure 6—video 3*). Minimum distance measures between the outer

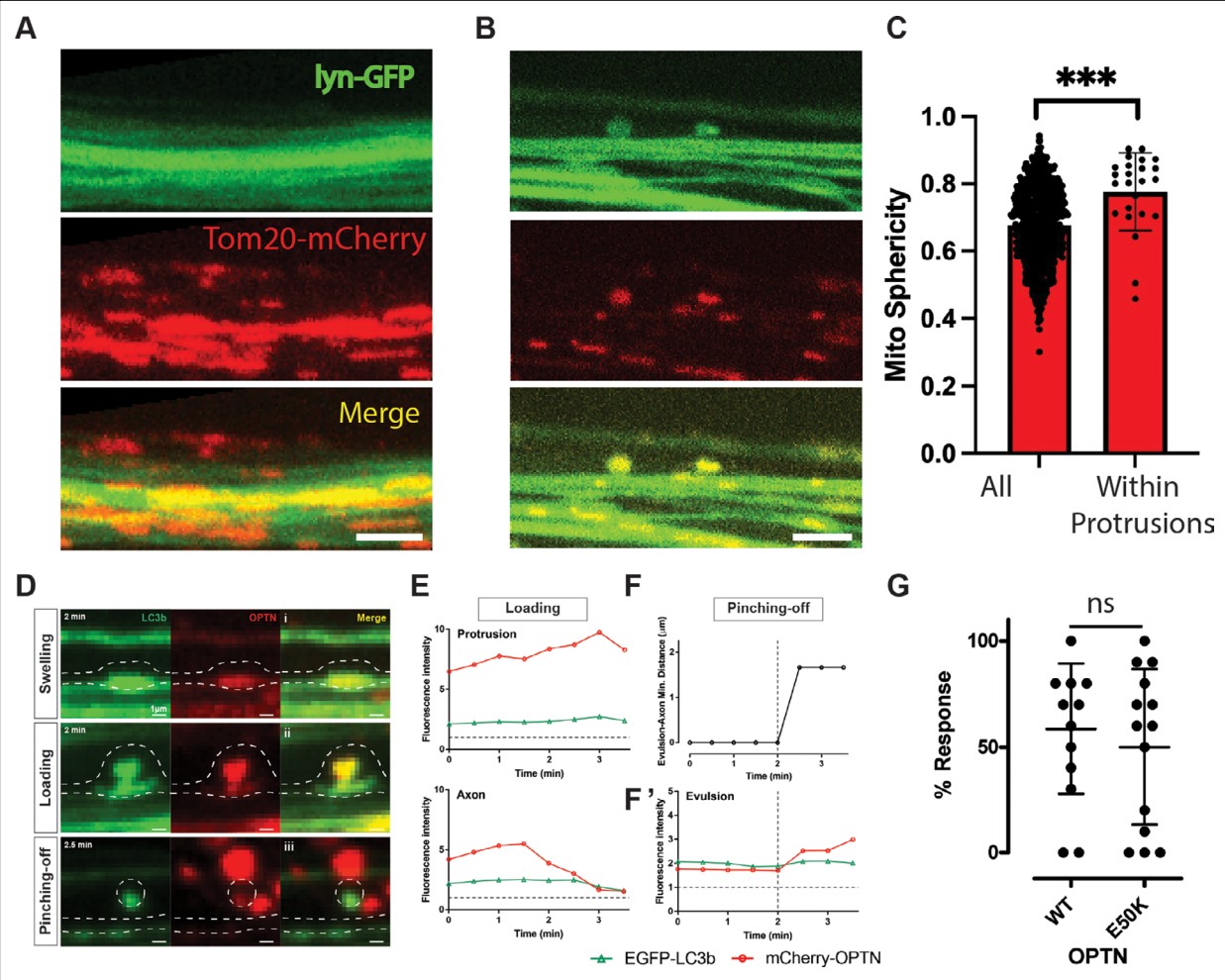

**Figure 6.** Mitochondria and OPTN leave retinal ganglion cell (RGC) axons through focal dystrophies. (**A**) Tom20-mCherry-labeled RGC mitochondria from sparse axons are found outside the axons, as labeled by a membrane EGFP transgene. Image is a z-projection spanning 4 µm, as external mitochondria have a diffuse pattern. (**B**) Axolemmal protrusions in sparse axons containing mitochondria. Image represents a single z plane. (**C**) Sphericity measures show that mitochondria within axolemmal protrusions differ from other axonal mitochondria. (**D**) Examples of axonal dystrophies in sparsely labeled *Tg(Isl2b:mCherry-OPTN(M98K)_ EGFP-LC3b)* axons containing partially co-localized stopped OPTN and LC3b; presented as a pseudo-sequence consistent with extra-axonal OPTN being the product of transcellular mitochondrial degradation. (i) Swelling, where the mitophagy machinery OPTN and LC3b is stopped together and focally enriched in a symmetric axonal dystrophy. (ii) Loading, where focally accumulated OPTN is enriched in an asymmetric axonal dystrophy. (iii) Pinching-off, where an axon evulsion is acutely separated from the axon of origin and appears to coalesce with other extra-axonal OPTN presumably within acidified organelles of a phagocyte. Scale bar, 1 µm. (**E**) Changes in fluorescence within the axon protrusion and directly beneath it within the axon of origin suggest an active loading process. (**F**) Minimum distance and (**F'**) changes in fluorescence within the axon evulsion show further increase in OPTN soon after separation from the axon. Gray dotted horizontal lines represent the mean fluorescence in the same axon away from the axon dystrophy, set at 1; thus, all fluorescence values shown represent focal concentrations of both LC3b and OPTN signal, suggestive of these being sites of axonal mitophagy. (**G**) Dot-avoidance response of animals expressing OPTN Wt or E50K transgenes. Not significant (ns); ***p < 0.0001 .

The online version of this article includes the following video(s) for figure 6:

**Figure 6—video 1.** Loading of OPTN into axonal dystrophy.
https://elifesciences.org/articles/103844/figures#fig6video1

**Figure 6—video 2.** Extruded mitochondria are static.
https://elifesciences.org/articles/103844/figures#fig6video2

**Figure 6—video 3.** Dystrophy pinches off from axon.
https://elifesciences.org/articles/103844/figures#fig6video3

boundaries of protrusion/evulsion and axon of origin (*Figure 6F*) and measures of the distance from its centroid to the nearest position along the axon (not shown) demonstrate that the detachment process is acute, occurring within a 30-s interval. The shape, size, and dynamics in these focal asymmetric axonal dystrophies lead us to conclude that at least some of the OPTN and mitochondria leave the axons by the transcellular degradation process we first described in the mouse optic nerve head (*Davis et al., 2014*). Finally, since the E50K OPTN mutation causes glaucoma, we then asked whether increased transcellular degradation might be associated with vision loss. For this, we used a behavior assay that reports RGC innervation of the optic tectum (*Dong et al., 2009*), the main brain target for RGC axons and the equivalent of the mouse superior colliculus. Animals expressing Wt or E50K OPTN showed no measurable visual impairment 3 days after inducing transgene expression with doxycycline, the time at which E50K OPTN expression results in an increase in transcellular degradation of mitochondria.

## Discussion

There is evidence that damaged axonal mitochondria are delivered back to or near the soma for degradation through OPTN-mediated mitophagy, although these studies were conducted mainly in cultured cells (*Moore and Holzbaur, 2016*; *Evans and Holzbaur, 2020*). Our studies in vivo are not inconsistent with this, as we did not image the retina where such degradation would be occurring; however, here we demonstrate that at least in tadpole optic nerves, large numbers of RGC axonal mitochondria are degraded within the optic nerve. We found that after expression of various mutated OPTNs, there are significant increases in the amount of OPTN and LC3b stopped within axons, and, in the one case we analyzed in most detail, the glaucoma-associated E50K mutation, also significant increases in the amount of stopped mitochondria, much of which is co-localized with OPTN. Arresting mitochondrial motility, which is mainly driven by degradation of the motor-adaptor protein Miro, is believed to be an early stage of mitophagy where dysfunctional mitochondria are stopped and sequestered for subsequent mitophagy through PINK1/Parkin-mediated pathway in both in vivo and in vitro settings (*Wang et al., 2011*; *Liu et al., 2012*; *Lovas and Wang, 2013*; *Ashrafi et al., 2014*; *Hsieh et al., 2016*). Since we demonstrate that OPTN and mitochondria co-localize within the optic nerve, our data strongly support the view that some steps associated with mitophagy occur locally in RGC axons far from the cell body. We found that under basal conditions, about half of axonal mitochondria are stopped; many of these have an elongated morphology and likely represent those mitochondria providing energy and other functions locally needed to support diverse cellular processes, including but not limited to action potential regeneration. However, after perturbations as slight as the injection of a balanced salt solution into the eye, the population of axonal mitochondria that are stopped within axons increases. We believe that these are unlikely to be more mitochondria supporting functions normally associated with axonal mitochondria, but rather that this second stopped population of axonal mitochondria is undergoing a process that at the very least shares some molecular machinery with mitophagy. However, this process does not appear to be conventional axonal mitophagy because neither the number of axonal mitochondria nor the fraction of stopped mitochondria is affected by chloroquine treatment, and, most importantly, because the majority of axonal mitochondria appear to be degraded by a transcellular degradation process, likely the same one that we previously described in the optic nerve head and cerebral cortex of normal mice (*Davis et al., 2014*).

The glaucoma-associated OPTN mutations not only had the largest fraction of stopped OPTN and LC3b within axons, but also led to large amounts of OPTN being found outside of the axons, some of which reached the surface of the optic nerve. Indeed, the studies based on sparse labeling of axons, which likely provide a more accurate estimate of the fraction of axonal mitochondria and OPTN outside of axons, suggest that the number of axonal mitochondria degraded outside of axons is large. After expression of Wt OPTN, which we believe represents the basal state as we also show that Wt OPTN expression has no effect on the behavior of axonal mitochondria, as much as 15.2 and 5.7% of the stopped mitochondria and OPTN, respectively, are outside of the axons, and in the case of expression of the E50K OPTN, those numbers increase to 35.8 and 21.8%. An independent measure that did not involve intravitreal injections, one based on a Tom20-mCherry transgene, provided a similar estimate of the fraction of axonal mitochondria outside of axons in the optic nerve, at 6.3%. However, all of these measures should be viewed as mere estimates, as the amounts of mitochondria outside of axons as determined by the Tom20-mCherry transgene differed some from those based on

Mitotracker labeling, especially when accounting for the mitochondria signal found on the surface of the optic nerve. It is likely that the Mitotracker signal outside of axons represents only mitochondria that are still morphologically intact, whereas the mCherry signal associated with the Tom20 or OPTN fusion constructs is expected to persist longer due to the intrinsic stability of the mCherry protein, indeed probably longer than most endogenous mitochondrial proteins. This may explain why, in the baseline state, including after expression of Wt OPTN, there is so little Mitotracker label on the surface of the optic nerve. Taking this reasoning one step further raises the interesting possibility that in our system, the various OPTN mutations may be acting not only by increasing the targeting of mitochondria for degradation within the axons, which they seem to do as we observe increased co-localization of mitochondria and OPTN even within moving populations within axons, but also that the OPTN mutants may separately affect the degradation of mitochondria. Indeed, there is evidence that OPTN may participate in multiple stages of mitochondria degradation in addition to bridging ubiquitinated mitochondria and LC3b, including the beginning steps of autophagosome formation (*Song et al., 2018*; *Evans and Holzbaur, 2020*) but also, through its interaction with Myosin-VI, also the ultimate delivery of these cargo to lysosomes (*Hu et al., 2019*). In our system, those later steps required for organelle degradation occur not within the RGC themselves but rather within astrocytes. Of note, in a co-culture model using ES-derived human RGCs and astrocytes, the E50K mutation in astrocytes was sufficient to cause neuronal dysfunction (*Gomes et al., 2022*). Thus, some or all of the pathological activity of glaucoma-associated OPTN mutations may occur outside of the RGCs themselves.

Mitochondria and OPTN found outside of axons could have in principle been eliminated by any of the cells present in the optic nerve, either the astrocytes which constitute the major local population, other resident cells such as NG2-cells or microglia, or alternatively by invading myeloid cells, all of which are known to have phagocytic capacity. We had previously shown that in the frog optic nerve, it is astrocytes that are the major phagocytes that clear extensive amounts of axonal and myelin debris using well-conserved phagocytic machinery, at least during a developmental remodeling event (*Mills et al., 2015*). Correlated light EM studies of sparsely labeled axons here demonstrate that the majority of these axonal material, including mitochondria, is here too degraded by the astrocytes, either in their fine processes that interdigitate deep in the nerve parenchyma, or far from the axons within the soma of the astrocytes, which in tadpoles at this stage reside exclusively on the surface of the optic nerve. The finding of large amounts of axonal mitochondria and OPTN on the surface of the optic nerve in astrocytes with prominent localization of an Aqp4-GFP reporter to their membranes is reminiscent of the glymphatic-like system that has been described in the mouse optic nerve (*Wang et al., 2020*); thus, whether mitochondria debris is cleared by a glymphatic pathway maybe should be investigated. Curiously, the axonal debris accumulating on the surface of the optic nerve was not uniformly distributed but rather was found mainly on the dorsal side. Whether this is related to anatomical asymmetries within the nerve or in the location of structures outside the nerve, such as lymph nodes, or whether this is in any way related to asymmetries in axonal degeneration observed in glaucoma, are all unknown but may be worthy of further exploration.

One important question is how the axonal mitochondria and OPTN reach the outside of axons. While we cannot exclude the possibility that some of the axonal mitochondria signal outside axons might derive from axons that have previously degenerated, the axonal structures live-imaged in this study are highly similar in size and shape to the mitochondria-filled protrusions still attached to axons and mitochondria-filled evulsions separated from axons that we had described in the optic nerve head of wild-type mice through both a mitochondria-targeted tandem EGFP-mCherry reporter and SBEM (*Davis et al., 2014*). Here, we provide evidence that one of twenty stopped axonal mitochondria, which is approximately one in fifty of all mitochondria within axons, is contained within a membranous protrusion, and the number that have recently pinched off is likely far higher.

Notably, the focal dystrophies which we observe on RGC axons are similar to what others refer to as exophers, both in *Caenorhabditis elegans* (*Melentijevic et al., 2017*) and in mammalian cardiac tissue (*Nicolás-Ávila et al., 2020*). In the case of exophers, the cellular extrusions occur constitutively, increase in response to various stressors, and contain diverse protein aggregates and dysfunctional organelles, which can but do not necessarily include mitochondria. In our case too, not all of the axonal protrusions contain mitochondria, but in those that did, the mitochondria were more spherical than the average axonal mitochondria, consistent with them being the product of mitochondria fission, quite possibly due to being damaged or dysfunctional. It may be that the other axonal protrusions that did

not contain mitochondria contained other organelles or protein aggregates, though the possibility that the outpocketings form prior to mitochondria entering them cannot be formally excluded. In *C. elegans*, after the pinching off of the exophers, they are processed to smaller 'starry night' vesicular structures that likely represent the endolysosomal intermediates on the path to the eventual degradation by lysosomes (*Wang et al., 2023*); it seems likely that the puncta containing axonal material that we observed within astrocyte processes and soma both at the light (e.g., *Figure 6—video 3*) and at the EM level likely are the equivalent of these 'starry night'. Thus, we suggest that axonal transcellular degradation of mitochondria, which we previously had also referred to as transmitophagy, is but an axonal variant of exopher generation that may have evolved as an alternative mechanism for long projection neurons such as RGCs to deal with more dysfunctional mitochondria or aggregates in distal regions of axons than could be dealt with by the more conventional cell-autonomous processes of somal or axonal mitophagy. What controls the balance between the cell-autonomous and non-cell-autonomous axonal degradation mechanisms remains an open question, although the control of mitochondria stalling, perhaps through the regulation of Miro stability by the PINK/Parkin pathway (*Wang et al., 2011*; *Liu et al., 2012*; *Lovas and Wang, 2013*; *Ashrafi et al., 2014*; *Hsieh et al., 2016*), is a possibility worthy of further study.

While the OPTN mutations that result in maximal transcellular degradation of RGC axon mitochondria are the very same mutations that in humans cause glaucoma, our study does not address whether increased transcellular degradation may in any way be relevant to the RGC axonal degeneration that defines this disease. At the time that our live-imaging studies were carried out, the tadpoles expressing the glaucoma-associated OPTN mutation did not show any obvious vision impairment, though that was after only 3 days after first inducing transgene expression. Whether more prolonged expression of these OPTN mutants will affect vision is yet to be determined. If they do not, it may be hard to interpret, as tadpoles of this stage have the ability to regenerate their axons and regain vision within days of all their axons being damaged (*Fague and Marsh-Armstrong, 2023*). However, a recent study in mouse optic nerve head (*Zhu et al., 2023*) where axonal mitochondria were labeled by Mitotracker retrograde transport found that the axonal Mitotracker signal within astrocytes increased in mice where intraocular pressure was elevated by microbead injection into the anterior chamber, an experimental paradigm used to model glaucoma. Further, they found that the Mitotracker signal was found largely within the distended processes of astrocytes that express high levels of Lgals3, the gene whose expression in astrocytes initially led us to the discovery of the transcellular degradation process, and which we had also shown increases in a different glaucoma model, DBA2/J mice (*Nguyen et al., 2011*). If an increase in transcellular axonal mitochondria degradation does contribute to glaucoma, a likely pathogenic mechanism might be inflammation resultant from the accumulation of immunogenic mitochondria debris outside of the axons, as that was shown to result when the clearance of mammalian cardiac exophers was blocked by removal of a phagocytic receptor (*Nicolás-Ávila et al., 2020*). Regardless of whether or not transcellular degradation of mitochondria contributes to the etiology of glaucoma, such a system might be a mechanism to handle the accumulation of detrimental aggregates and damaged organelles within diverse axons, which would be of obvious relevance to a variety of neurodegenerative disorders that affect axonal biology early in their progression.

## Materials and methods
### Transgenes

- pCS2(Isl2b):rtTA2_pTRETightBI_mCherry-OPTNs_EGFP-LC3b: mCherry flanked by additional restriction sites was amplified from pTRETightBI-RY-0 (Addgene plasmid # 31463; http://n2t.net/addgene:31463; RRID:Addgene_31463) (*Mukherji et al., 2011*) using CD160 and CD161 and recloned into XmaI and EcoRV sites of the same vector. Full-length human OPTN, from the Mammalian Gene Collection (CloneID 3457195, purchased from OpenBiosystems) was amplified with CD162 and CD163 and placed into MluI and EcoRV sites, creating pTRETightBI:mCherry-OPTN_NLS-EYFP. The β-globin polyadenylation sequence was then amplified from AAV2:MitoEGFPmCherry (*Davis et al., 2014*) using CD60 and CD61 and inserted in between EcoRV and AatII sites downstream of OPTN. To make the EGFP-LC3b part of the constructs, the Egfp was first amplified from pCS2:mito-EGFP-mCherry (*Davis et al., 2014*) using CD53 and CD91h and cloned into pTRETight:MitoTimer (Addgene plasmid # 50547; http://n2t.net/addgene:50547; RRID:Addgene_50547) (*Hernandez et al., 2013*), with EcoRI and XbaI, creating the

construct pTRETight:EGFP. Then, full-length mouse LC3b from OpenBiosystems (CloneID 5319360) was amplified with CD58 and CD59 and cloned into SalI-NheI sites of this construct in order to make pTRETight:EGFP-LC3b. Then, EGFP-LC3b was moved into the previously created pTRETightBI:mCherry-OPTN_NLS-EYFP by cloning it into the EcoRI and XbaI sites, thus replacing the NLS-EYFP. In order to put both EGFP-LC3b and mCherry-OPTN under inducible control, the first step was to create a version of pCS2:rtTA2, previously shown to have low baseline and high inducibility in transgenic *X. laevis* (*Das and Brown, 2004*; *Mills et al., 2015*), but with an additional multiple cloning site to facilitate assembly. The rtTA2 itself was amplified from Blbp:rtTA2 (*Mills et al., 2015*) using CD158 and CD159, and cloned into PmeI and PacI sites of a version of pCS2 vector where the CMV promoter had been previously replaced by 600 bp upstream sequences of the Cardiac Actin promoter (as well as a multiple cloning site or MCS containing ZraI, AatII, PmeI, and PacI restriction sites), accomplished by amplifying the promoter itself from *X. laevis* genomic DNA using CD156 and CD157. Then, the majority of pTRETightBI:mCherry-OPTN_EGFP-LC3b, excluding the backbone sequences, was placed between the AatII-NotI sites of this pCS2(MCS-CA600):rtTA2, before removing the CA600 promoter itself by digestion with PacI and NotI followed by blunting and self-ligation. pCS2(MCS):rtTA2_pTRETightBI:mCherry-OPTN_EGFP-LC3b was then cut with SalI and ZraI, and most of it moved into the SmaI and XhoI sites of a derivative of the previous pCS2 (1 kb Isl2b) vector containing both 1 kb of the RGC-specific Isl2b promoter as well a Flip-recombinase excisable Kanamycin antibiotic selection cassette (pCS2(1 kb Isl2b):GFP3-Frt-Kan-Frt; *Watson et al., 2012*), into which a SmaI restriction site had been previously added immediately down-stream of HindIII, so as to generate pCS2(1 kb Isl2b):rtTA2_pTRETightBI:mCherry-OPTN_EGFP-LC3b_FKF. All the OPTN mutations (E50K, M98K, F178A, D474N, E478G, and H486R) were then introduced into this construct by QuikChange Site-Directed Mutagenesis Kit (Stratagene) using the corresponding primers (Primer List below, mutagenized bases are shown in upper-case) and verified by Sanger sequencing. The final step of placing all these under control of the full 20 kb of the zebrafish Isl2b promoter, which drives RGC-specific expression in both zebrafish (*Pittman et al., 2008*) and *X. laevis* (*Watson et al., 2012*), was accomplished by recombineering as previously described (*Watson et al., 2012*).

- pCS2(Isl2b:Tom20-mCherry-Apex2_msSOD2UTR): First, to improve the mitochondria targeting efficiency of our original mitochondria transcellular degradation reporter, pCS2(1 kb Isl2b):Mito-EGFP-mCherry (*Davis et al., 2014*), the SV40pA was replaced by the 3'UTR of msSOD2 (*Kaltim-bacher et al., 2006*) amplified from mouse brain cDNA (Clontech 637301) using primers AA35 and AA36 and inserted into the XbaI and NotI sites. Then, the mitochondria targeting sequence from Cox8 in this construct was replaced first by Sncg amplified from the same mouse brain cDNA with AA41 and AA42, using HindIII and BamHI, and from there replaced again by the mouse Tom20 sequences amplified from the same mouse brain cDNA with AA47 and AA48, using XmaI and BamHI. To make the Tom20-linker-mCherry-linker-FlagApex2-msSOD2UTR, all relevant fragments were combined in a four-way Gibson Assembly using the following primers: CD410 and CD621 (Fragment 1), CD620 and CD622 (Fragment 2), CD623 and CD627 (Fragment 3), and CD626 and CD411 (Fragment 4). This construct was originally made to contain within the first fragment a *Xenopus* Rhodopsin promoter (*xop*; *Zhang et al., 2008*), but then the entire cDNA along with the polyadenylation sequence was moved with HindIII and NotI into a construct with 1 kb of the zebrafish-derived Isl2b promoter and Kanamycin selection cassette, so as to make the final construct, referred to simply as Isl2b:Tom20-mcherry in the text, by recombineering as described above.

- pCS2(xtFapb7):mTagBFP2-Ras: BFP from pBAD-mTagBFP2 (Addgene plasmid # 34632; http://n2t.net/addgene:34632; RRID:Addgene_34632) (*Subach et al., 2011*), was amplified using FK99 and FK98 and cloned into HindIII and BglII sites of a pCS2 vector. The oligonucleotides encoding BglII-Ras farnesylation sequence (AAGCTGAACCCTCCTGATGAGAGTGGCCCCGG CTGCATGAGCTGCAAGTGTGTGCTCTCCTGA) -XbaI was inserted into BglII and XbaI sites of the vector by using complementary oligonucleotides. Then the full cDNA of mTagBFP2-Ras and SV40 polyadenylation sequence was placed in between HindIII and NotI sites of pCS2(xtFabp7) vector.

- pCS2(xtFapb7):xlAqp4-GFP: A Aqp4 cDNA was amplified from tadpole optic nerve mRNA using primers NMD18 and NMD20 and cloned into HindIII and NheI sites of a pCS2(xtFabp7) construct (*Mills et al., 2015*) with a unique NheI site in frame upstream of GFP. This DNA was then used as template for PCR reactions using NMD 29 and CD411, and NMD30 and CD410. In order to mutagenize the first stop codon of Aqp4 into a tryptophan through a Gibson ligation of the two amplicons.

- pCS2(xtFapb7):EGFP-mCherry-Lc3b: First, pCS2(xtFapb7):EGFP-Lc3b was made by amplifying the same mouse Lc3b as above with EM284 and EM285 and after digestion with NheI and XhoI cloning it into a construct carrying the same pCS2(xtFabp7) backbone as above digested with the same enzymes. Then, the EGFP-mCherry cassette was amplified from Mito-EGFP-mCherry (*Davis et al., 2014*) using EM232 and EM339 and after digestion with HindIII and XbaI cloned into the HindIII and SpeI sites of pCS2(xtFapb7): EGFP-LC3b.

| ID | Sequence | Gene | Forward/Reverse |
|---|---|---|---|
| AA 35 | acgacgtctagaactcacggccacattgagtg | msSOD2UTR | Forward |
| AA 36 | acgacggcggccgctggtgtactgtgaaactgtgacc | msSOD2UTR | Reverse |
| AA 41 | acgacgaagcttcccggggccaccatggacgtcttcaagaaaggcttc | msSncg | Forward |
| AA 42 | acgacgggatccactagtgtcttctccactcttggcctctt | msSncg | Reverse |
| AA 47 | acgacgcccggggccaccatggtggggccggaacagcgcc | msTom20 | Forward |
| AA 48 | acgacgggatccttccacatcatcttcagccaagct | msTom20 | Reverse |
| CD 19 | gcagatgaaagagctcctgaccAAGaaccaccagctgaaagaagc | huOptn E50K | Forward |
| CD 20 | gcttctttcagctggtggttCTTggtcaggagctctttcatctgc | huOptn E50K | Reverse |
| CD 21 | ctgttctgattttcatgctGGAagagcagcgagagagaaaattc | huOptn E478G | Forward |
| CD 22 | gaattttctctctcgctgctctTCCagcatgaaaatcagaacag | huOptn E478G | Reverse |
| CD 23 | gcggctcctcagaagattccGCTgttgaaattaggatggc | huOptn F178A | Forward |
| CD 24 | gccatcctaatttcaacAGCggaatcttctgaggagccgc | huOptn F178A | Reverse |
| CD 25 | gcagcgagagagaaaattCGTgaggaaaaggagcaactggc | huOptn H486R | Forward |
| CD 26 | gccagttgctccttttcctcACGaattttctctctcgctgc | huOptn H486R | Reverse |
| CD 53 | acgacggaattcaagcttgccaccatggtgagcaagggcgaggagctg | EcoRI-HindIII-Kozak-Egfp | Forward |
| CD 58 | acgacggctagcccgtccgagaagaccttcaagc | NheI-msLC3b | Forward |
| CD 59 | acgacggtcgacttacacagccattgctgtcccg | msLC3b-SalI | Reverse |
| CD 60 | acgacggtcgacgatatcgctagcctgaggatccgatcttttttccc | SalI-EcoRV-NheI-βGlobin pA | Forward |
| CD 61 | acgacggacgtctagggataacagggtaatgcggccgcc ttccgagtgagagacacaaaa | AatII-ISceI-NotI- βGlobin pA | Reverse |
| CD 91h | acgacgtctagagtcgacttaattaagctagccttgtacagctcgtccatgccga | Egfp-NheI-PacI-SalI-XbaI | Reverse |
| CD 156 | acggtcgacgacgtcacctggtgtttaaacttaa ttaatccactgcattctgtcctgaga | SalI-AatII-DraIII-SexAI-PmeI-PacI-CA promoter 600 | Forward |
| CD 157 | acgacgaagctttggggactgagctgtcaatttata | HindIII-CA promoter 600 | Reverse |
| CD 158 | acgacggtttaaacgccaccatgtctagactggacaagagcaaag | PmeI-rtTA2 | Forward |
| CD 159 | acgacgttaattaagaattaaaaaaacctcccacacctc | PacI-SV40pA | Reverse |
| CD 160 | acgacgcccggggccaccatggtgagcaagggcgaggaggat | XmaI-Kozak-ATG-mCherry | Forward |
| CD 161 | acgacggatatccgtcgtacgcgtcttgtacagctcgtccatgccg | mCherry-noSTOP-MluI-EcoRV | Reverse |
| CD 162 | acgacgacgcgttcccatcaacctctcagctgcc | MluI-noATG-huOPTN | Forward |
| CD 163 | acgacggatatcctaaatgatgcaatccatcacgtgaatc | huOPTN-EcoRV | Reverse |
| CD 177 | agaagcaaaagagcgtctaAAGgccttgagtcatgagaatg | huOptn M98K | Forward |
| CD 178 | cattctcatgactcaaggcCTTtagacgctcttttgcttct | huOptn M98K | Reverse |
| CD 265 | agatggaagtttactgttctAATttcatgctgaaagagca | huOptn D474N | Forward |
| CD 266 | tgctctttcagcatgaaaATTagaacagtaaacttccatct | huOptn D474N | Reverse |
| CD 410 | ggcatcgtggtgtcacgctcg | AmpR | Forward |
| CD 411 | cgagcgtgacaccacgatgcc | AmpR | Reverse |
| CD 620 | ggacctggaagtattgctaccagaattcaaatggtgagcaagggcgaggaggat | Tom20-linker-mCherry | Forward |
| CD 621 | atcctcctcgcccttgctcaccatttgaattctggtagcaatacttccaggtcc | Tom20-linker-mCherry | Reverse |

*Continued on next page*

*Continued*

| ID | Sequence | Gene | Forward/Reverse |
|---|---|---|---|
| CD 622 | gctagcgcctccgccagagcctccgcctccacgcgtcttgtacagctcgtccatgccg | mCherry- Glycine- linker | Reverse |
| CD 623 | acgcgtggaggcggaggctctggcggaggcgctagcatggactacaaggatgacgacg | Glycine-linker- FlagAPEX2 | Forward |
| CD 626 | ggagcgcctgaccctggactaactcgagcctctagaactcacggccacatt | APEX2 msSOD2UTR | Forward |
| CD 627 | aatgtggccgtgagttctagaggctcgagttagtccagggtcaggcgctcc | APEX2 msSOD2UTR | Reverse |
| EM 232 | aaggagaagcttgccaccatggtgagcaagggcgaggagc | EGFP-mCherry | Forward |
| EM 284 | aaggaggctagcccgtccgagaagaccttcaag | msLC3b | Forward |
| EM 285 | aaggagctcgagttacacagccattgctgtccc | msLC3b | Reverse |
| EM 339 | aaggagtctagacttgtacagctcgtccatgc | EGFP-mCherry | Reverse |
| FK 99 | aaggagaagcttgccaccatgagcgagctgattaaggag | Kozak-HindIII- BFP | Forward |
| FK 98 | acgacgagatctatttagcttgtgccccagtttgctaggg | BFP-BglII | Reverse |
| NMD 18 | aaggagaagcttgccaccatggtggcatgtaaaggagtctgg | HindIII-xlAqp4 | Forward |
| NMD 19 | aaggagctcgagcaatatatctgtctcttttactggaag | XhoI-xlAqp4 (no stop 2) | Reverse |
| NMD 20 | aaggaggctagccaatatatctgtctcttttactggaag | NheI-xlAqp4 (no stop 2) | Reverse |
| NMD 29 | ggtattatcttcggtatgGctagaagaaagaaatg | Aqp4 stop ->Ala | Forward |
| NMD 30 | catttctttcttctagCcataccgaagataatacc | Aqp4 stop ->Ala | Reverse |

## Transgenesis and chloroquine treatment

Transgenic *X. laevis* were generated by REMI transgenesis (*Amaya and Kroll, 1999*) using optimizations for larger DNA constructs that have been described previously (*Mills et al., 2015*). In all cases, DNAs were linearized overnight with NotI in the injection buffer, followed by 65°C 20-min inactivation of the enzyme. Embryos were kept in 0.1X Marc's Modified Ringer's (MMR) with 50 µg/ml of gentamicin in sterile glass Petri dishes for the first 2 days, the first day at 16°C and the second at room temperature, and then subsequently kept in 0.1X MMR in glass bowls at room temperature and under a 12/12 light/dark cycle.

Animals were treated with 100 µM chloroquine diphosphate dissolved in 0.1X MMR for either 10 hr in the case of the measurements of autophagosomes with the *pCS2(xtFapb7):LC3b-EGFP-mCherry* transgene, and for 24 hr in the case of measurements of mitochondria movement within axons and amount of mitochondria signal outside axons.

## Generation of optic nerves with sparsely labeled axons

Approximately 10–50 eye anlage cells were surgically transferred from *X. laevis* homozygous or heterozygous for Isl2b:mCherry-OPTN(Wt/E50K/M98K)_EGFP-LC3b into either wild-type or *Tg(xt-Fabp7:mTagBFP2-Ras)* hosts at a developmental stage prior to retinal cell differentiation (at between NF stage 24 and 32). Surgeries were carried under Zeiss Stemi 2000 stereomicroscopes using handheld curved tungsten micropins on animals anesthetized with 0.2 g/l of Tricaine Methanesulfonate (MS-222) in filter sterilized 0.1X MMR containing 50 µg/ml of gentamicin and with additional buffering provided by 2 mM HEPES, pH 7.4–7.6, in 35 mm Petri dishes. Animals were immobilized during surgeries in disposable clay-mold (Permoplast, non-toxic) inserts with adjacent grooves created with a pipette tip and then altered with forceps to loosely anchor donors and hosts in parallel near each other. At approximately 10 min after transplantation, the animals were released from their clay-mold enclosures and donors and hosts returned together to the same dish containing fresh sterile-filtered 0.1X MMR 2 mM HEPES with 50 µg/ml of gentamicin. Successful transplants, which were approximately half of the number of operations performed, were defined as those with donors confirmed as transgenic (since operations are done prior to onset of transgene expression), and whose hosts had eyes of normal size and only a few RGC axons labeled, as primarily assessed under a Leica MZ10F fluorescence-equipped stereomicroscope and then verified through confocal live-imaging (see details below).

## Intravitreal injection of mitotracker

Mitotracker Deep Red is a far-red fluorescent dye that chemically stains mitochondria in live cells (*Audano et al., 2021*; *Weiss-Sadan et al., 2019*). Fresh 200 µM of MitoTracker Deep Red FM (Cell Signaling Technology, #8778) dye solution was prepared by diluting a 5 mM of stock solution prepared in DMSO with filtered 0.5X MMR on the day of injection. The Mitotracker solution was front-loaded into a borosilicate glass capillary (World Precision Instruments) that had been pulled and then broken to 1–2 µm tip diameter using fine forceps, and then injected intravitreally into left eyes of anesthetized animals at around NF stage 48 (*Nieuwkoop and Faber, 1994*), using a pressure injector (Narishige IM-300 Microinjector); successful injections were verified by observing an acute but mild swelling of the eye at the time of injection. After injection, the animals were returned to a new dish containing 0.1X MMR at room temperature and imaged 3.5 hr after injection for *Figure 1—figure supplement 1A, B* and 15–18 hr after injection for all the other experiments.

## Optic nerve live imaging

In the case of doxycycline-inducible transgenes, *X. laevis* tadpoles were placed in 0.1X MMR with 50 µg/ml of doxycycline for 2–3 days before imaging. At around NF stage 48, *X. laevis* tadpoles were anesthetized in 0.1X MMR with 0.2 g/l of MS-222 and mounted into custom Sylgard 184 silicone elastomer (World Precision Instruments) molds. Molds were created using glutaraldehyde-fixed tadpoles at the same stages slightly angled using insect pin inserts so as to make left optic nerves parallel to the imaging plane, and then modified with surgical blades to provide an opening before the mouth to facilitate breathing. Tadpoles immobilized in the molds were weighed down by an 18 mm circular coverslip and imaged from beneath in a 35 mm dish modified with a 22 mm square glass coverslip as bottom. Optic nerves of the animals were imaged by spinning disk confocal microscopy (Dragonfly 503 multimodal imaging system, Andor Technology, Belfast, UK) with a ×40/1.10 (magnification/ numerical aperture) HC PL APO water immersion objective, using a Leica DMi8 inverted microscope (Leica, Wetzlar, Germany), an iXon Ultra 888 EMCCD camera (Andor Technology, Belfast, UK), Fusion Software (Andor Technology, Belfast, UK) and laser lines of 100 mW 405 nm, 50 mW 488 nm, 50 mW 561 nm, and 100 mW 643 nm; first, a time-series was obtained at a single focal plane imaged for 1 min at 1 Hz, and the entire thickness of the nerve at that same location was z-scanned at 1 µm steps. Videos at 7 fps and images were created using FIJI ImageJ.

For some sparsely labeled axons, 5–10 µm thick regions of the optic nerve were subjected to repetitive z-scan live-imaging either at 30-s intervals for 5 min or at 2-min intervals for 10 min. Acquired images were saved as separate image sequence files using FIJI ImageJ and imported into IPLab software (Scanalytics). Regions of interest sampled at different time points were then registered and quantified using a macro written in IPLab. All the relevant videos (3 fps for M98K and 1 fps for E50K videos) and images were created using FIJI ImageJ.

## Kymograph analyses

Axonal movements of mitochondria, OPTN and LC3b: acquired time-series images were saved as separate image sequence files in FIJI ImageJ after being processed by HyperStackReg or StackReg Plugin to eliminate object drift. The aligned image sequence files were restacked using IPLab software, followed by either manual (for the sparsely labeled axons and curved axons) or semi-automatic tracing using IPLab macros so as to visualize and measure fluorescence intensity changes over time in the form of kymographs. For the semi-automatic tracings, the outer contour of the optic nerve was traced, and the approximately 30 µm diameter nerve was automatically divided into 0.9–1.8 µm parallel swaths, which contained small stretches of axons running in parallel; using such sampling distance and thickness, the same moving objects within axons were not multiply counted, but some of the larger immotile objects might have been multiply counted. Since there was high concordance between ratios of moving and static objects and the velocities of the moving objects when comparing whole-nerve semi-automatic kymographs and the more conventional single axon tracing used in the sparse labeling experiments, any error introduced by the novel semi-automatic tracing of axons was deemed minimal. Axonal movement was determined by counting the percentage of moving and stopped objects and measuring the speed of moving over the 1 min imaging period. Objects were classified as moving if their velocity was 0.1 µm/s or faster. Ratios and average speeds were calculated in Excel.

Co-localization measures: The same aligned and stacked images from the analysis of axonal movements were reprocessed and analyzed by a separate co-localization script but by the same logic through IPLab as described previously, except displaying multiple-channel fluorescence images along the merged color images, and sequentially tracing objects labeled by multiple fluorophores first, followed by those labeled by individual fluorophores. Ratios and average speeds were calculated in Excel.

## Nerve 3D reconstruction and quantification in Imaris

Z-scan confocal images of optic nerves expressing mCherry-OPTN and EGFP-LC3b were imported into Imaris software (Bitplane). Because LC3b is localized not only in autophagosomal membrane, but also is observed more diffusely within cytosol, including axoplasm (*Fu et al., 2014*; *Wong and Holzbaur, 2014b*; *Tammineni et al., 2017*; *Stavoe et al., 2019*; *Evans and Holzbaur, 2020*; *Kuijpers et al., 2021*; *Boecker et al., 2021*), this EGFP-LC3b signal was used to create a mask to define the boundary of the RGC axons and to separately measure the mCherry-OPTN or far-red mitochondria signal that co-localized or not with the axons. The Imaris mask toolset was used to produce channels for OPTN, mitochondria, or OPTN–mitochondria either co-localizing or not with LC3b. The % outside axon values were calculated in Excel by comparing the volumes of objects outside of the LC3b mask relative to the total volumes.

## Correlated light SBEM

After optic nerve live imaging, tadpoles were fixed overnight with 4% paraformaldehyde, re-imaged for confocal imaging and positioning, then fixed overnight with 2% EM grade glutaraldehyde (18426, Ted Pella Incorporated) in 0.15 M sodium cacodylate buffer (SCB) containing 2 mM $CaCl_2$. The tadpoles were then prepared for serial block-face imaging (SBEM) as described previously (*Deerinck et al., 2010*). Briefly, tadpoles were washed with SCB then incubated in 2% $OsO_4$ + 1.5% potassium ferrocyanide in SCB containing 2 mM $CaCl_2$ for 1 hr. Tadpoles were washed with double distilled water ($ddH_2O$) and incubated in 0.05% thiocarbohydrazide for 30 min. Tadpoles were then washed again with $ddH_2O$ and stained with 2% aqueous $OsO_4$ for 30 min. Tadpoles were again washed in $ddH_2O$ and placed in 2% aqueous uranyl acetate overnight at 4°C. Tadpoles were washed again with $ddH_2O$ and stained with en bloc lead aspartate for 30 min at 60°C. After a last wash in $ddH_2O$, tadpoles were dehydrated on ice in 50%, 70%, 90%, and 100% ethanol solutions for 10 min each, followed by two 10-min incubations in dry acetone. Tadpoles were placed in 50:50 dry acetone/Durcopan resin overnight, followed by three changes of 100% Durcopan for ~12 hr each. Tadpoles were then allowed to harden in Durcopan at 60°C for 48 hr.

Tissue blocks were cut from hardened Durcopan, and the whole head of the tadpole was mounted on plastic dummy blocks, trimmed down to only the region surrounding the left optic nerve, eye, and brain. Exact location of optic nerve and the rest of the region of interest determined by X-ray microscopy imaging of tissue blocks performed on a Zeiss Xradia Versa 510 X-ray microscope (XRM) instrument (Zeiss X-Ray Microscopy, Pleasanton, CA, USA). XRM tilt series were generally collected at 120 kV and 10 W power (75 µA current) and used to create 3D reconstructions of the embedded tissue. SBEM data was collected with a 3View unit (Gatan, Inc, Pleasanton, CA, USA) installed on a Gemini field emission SEM (Carl Zeiss Microscopy, Jena, Germany). Volume was collected at 2.5 kV accelerating voltage, with a raster size of 20k × 17k and pixel dwell time of 1.0 µsec. The pixel size was 5.0 nm and section thickness was 60 nm. SBEM volumes were analyzed and annotated in 3Dmod (3dmod Version 4.11.24, Copyright (C) 1994–2021 by the Regents of the University of Colorado).

## Behavioral assay of vision

Dot avoidance behavior was carried out essentially as previously described (*Fague and Marsh-Armstrong, 2023*). Ten informative trials were performed for each tadpole; that is, trials where the tadpoles might have initiated or changed movement, but it is unclear whether that was in response to the oncoming dot were not counted. A 50% response rate is characteristic of normal vision.

## Statistics

Statistical analyses involved the comparison of means using unpaired, two-tailed Student's *t*-tests or two-way ANOVAs following Tukey's post hoc test for multiple comparisons using GraphPad Prism

software. All the bar graphs represent the means ± SEM, where values were first averaged per animal and N is the number of animals. In the stacked bar graphs (except *Figure 2G*, *Figure 1—figure supplement 1E*, and *Figure 2—figure supplement 1H, I*) showing percentage of each movement, all statistical comparisons were performed using all the axonal movements including stationary, anterograde, and retrograde, although only the statistically significant comparisons between the stationary groups (against Wt OPTN or control group) are shown. In all the velocity graphs (except *Figure 1—figure supplement 1F*), all statistical comparisons were performed throughout all the transgenic lines and movements (both anterograde and retrograde movements), and only the statistically significant comparisons against the Wt OPTN or control group are shown.

All animal experiments were carried out in accordance with procedures approved by the Institutional Animal Care and Use Committee of University of California, Davis.

## Acknowledgements

This work was supported by the National Eye Institute (R01EY026471 and R01EY029087) (N M-A). Correlated light, XRM, and Volume EM analysis was performed at the National Center for Microscopy and Imaging Research, with support from NIH grants U24 NS120055, 1S10OD021784 and National Science Foundation – NSF2014862-UTA20-000890 (MHE). Deposition and management of Volume EM data within the Cell Image Library was further supported by NIH grant R01 GM82949 (MHE). The authors thank Elizabeth A Mills and Ferdinand Kaya for their initial work on generating the original Isl2b- and Fabp7-promoter constructs. Image analyses were performed through the use of UC Davis Health Sciences Advanced Imaging Facility supported by the NEI UC Davis Core grant (P30-EY012576).

## Additional information

### Funding

| Funder | Grant reference number | Author |
|---|---|---|
| National Eye Institute | EY026471 | Nicholas Marsh-Armstrong |
| National Eye Institute | EY029087 | Nicholas Marsh-Armstrong |
| National Eye Institute | P30-EY012576 | Nicholas Marsh-Armstrong |
| National Institutes of Health | U24 NS120055 | Nicholas Marsh-Armstrong |
| National Institutes of Health | ISI0OD021784 | Nicholas Marsh-Armstrong |
| National Institutes of Health | GM82949 | Nicholas Marsh-Armstrong |
| National Science Foundation | 2014862-UTA20-000 | Nicholas Marsh-Armstrong |
| National Institutes of Health | T32GM 153586 | Hector H Navarro |
| National Institutes of Health | T32GM13574 | Hector H Navarro |

The funders had no role in study design, data collection, and interpretation, or the decision to submit the work for publication.

### Author contributions

Yaeram Jeong, Data curation, Software, Formal analysis, Validation, Investigation, Visualization, Methodology, Writing - original draft; Chung-ha O Davis, Conceptualization, Investigation, Methodology, Writing – review and editing; Aaron M Muscarella, Formal analysis, Investigation, Visualization, Methodology; Hector H Navarro, Formal analysis, Validation, Investigation, Visualization; Viraj Deshpande, Formal analysis, Investigation, Visualization; Lucy G Moore, Formal analysis, Investigation; Keun-Young Kim, Investigation; Mark H Ellisman, Resources, Writing – review and editing; Nicholas

Marsh-Armstrong, Conceptualization, Resources, Data curation, Software, Formal analysis, Supervision, Funding acquisition, Validation, Investigation, Visualization, Methodology, Project administration, Writing – review and editing

### Author ORCIDs
Nicholas Marsh-Armstrong ⓘ https://orcid.org/0000-0002-7843-6651

### Ethics
The study was carried out in accordance with recommendations in the Guide for the Care and Use of Laboratory Animals of the National Institutes of Health. All animal work was carried out according to approved institutional care and use committee protocols (#22908) of the University of California Davis. Surgeries and imaging were carried out under MS-222 anesthesia and all efforts were made to minimize pain or distress to the animals.

### Decision letter and Author response
Decision letter https://doi.org/10.7554/eLife.103844.sa1
Author response https://doi.org/10.7554/eLife.103844.sa2

## Additional files

### Supplementary files
MDAR checklist

### Data availability
Data has been uploaded to Dryad (https://doi.org/10.5061/dryad.zkh1893nt).

The following dataset was generated:

| Author(s) | Year | Dataset title | Dataset URL | Database and Identifier |
|---|---|---|---|---|
| Marsh-Armstrong N, Jeong Y, Davis CO, Muscarella AM, Navarro HH, Deshpande V, Moore LG, Kim KY, Ellisman MH | 2025 | Glaucoma-associated optineurin mutations increase transcellular degradation of mitochondria in a vertebrate optic nerve | https://doi.org/10.5061/dryad.zkh1893nt | Dryad Digital Repository, 10.5061/dryad.zkh1893nt |

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
