## [Editor Report]

This work describes the effect of Optineurin (OPTN) mutations in the transcellular degradation of retinal ganglion cell mitochondria by astrocytes in the Optic Nerve, a process termed, "transcellular degradation of mitochondria". The authors perform compelling live imaging studies of the *Xenopus laevis* optic nerve to track neuronal mitochondrial movement and expulsion in an intact nervous system. These important findings demonstrate that Optineurin mutations that are associated with disease increase the stationary pool of mitochondria resulting in increased rates of transcellular degradation.

---

## [Decision Letter]

[Editors' note: this paper was reviewed by Review Commons.]

---

## [Author Response]

Reviewer #1 (Evidence, reproducibility and clarity (Required)):Glaucoma-associated optineurin mutations increase transmitophagy in vertebrate optic nerve.SummaryIn Jeong et al., the authors perform live imaging of the *X. laevis* optic nerve to track neuronal mitochondrial movement and expulsion in an intact nervous system. The authors observe similar mitochondrial dynamics in vivo as previously described in other systems. They find that stationary mitochondria are more likely to be associated with OPTN, suggestive of mitochondria undergoing mitophagy. Forced expression of OPTN mutations results in a larger pool of stationary mitochondria that colocalize withLC3B, and OPTN. Finally, the authors argue that extra-axonal mitochondria are observed more frequently in OPTN mutants, suggesting that mutations in OPTN that are associated with disease can lead to an increase in the expulsion of mitochondria through exopher-like structures.Major Findings and impact:The authors establish that mitochondria dynamics can be tracked in the *X. laevis* optic nerve.OPTN mutations increase the stationary pool of mitochondria and likely result in increased rates of mitophagy.Exopher-like structures containing mitochondria and LC3 can be expelled from the optic nerve and increase in the presence of OPTN mutations. These structures were observed in a living system and have interesting implications in the context of disease.Concerns:The authors state in their results that the secreted blebs are exophers. While these initial observations are consistent with exophers, additional data are needed to strengthen this claim. For example: what are the sizes of secreted vesicles? Do all express LC3? How frequently do these occur? From where are they expelling? Alternatively, the discussion of exophers could be moved to the discussion.

We agree that calling the axon shedding intermediates “exophers” was an overreach on our part. While we believe that in all probability time will demonstrate this to be the case, reviewers are correct in stating that putting our work in the context of exophers is best left to the discussion. We have removed all mention of exophers from the results and graphical abstract and now use the term only once in the discussion. We do provide detail as to the frequency of the structures, what fraction contain mitochondria, and morphological parameters of the contained mitochondria. And while all of these new data support them being exophers, the point remains that the use of the nomenclature “exopher” in the Results section was inappropriate.

Quantifications in sparse labeling experiments seem quite surprising and concerns related to these findings should be addressed. As the authors used LC3b expression to represent axonal volume, the authors should demonstrate that this is the case using an axonal fill or membrane marker in both the wt and E50K conditions. This is important as it is unclear whether LC3b expression is consistent between the wild type and the E50K conditions. Lower expression of LC3b in E50K could account for the large changes in axonal width that seem to be observed and could confound the measured amount of expelled mitochondria.

We agree that using EGFP-LC3b as a “cell fill” was problematic in a situation where the interventions likely perturb autophagy/mitophagy and therefore might have also perturbed LC3b. We do provide some axon width and LC3b-EGFP intensity data for a partial dataset that had been imaged side-by-side, showing that expression of LC3b is not different in the two conditions. We also provide independent measures of extraaxonal mitochondria based on a membrane-GFP reporter. While in principle there would be value to repeat the studies of Wt vs. E50K in the context of the membrane-GFP reporter, these experiments would involve new constructs and new breedings, and would likely take months to years to complete.

Could large amounts of exogenous mitochondria in explant experiments be from cells that died during the plantation?

The concern that some of the exogenous mitochondria signal might derive from degenerating axons is one that we worry much about, and not only in the transplantation experiments. In our sparse labeling experiments we do occasionally see axons undergoing Wallerian degeneration, but it is rare and does not appear to be more common in the expression of the mutated OPTN, at least not at the stage after transgene expression that the analyses were performed. We do provide new data that expression of E50K OPTN does not compromise vision at the time that experiments were carried out, ruling out that extra-axonal mitochondria are the result of large-scale degeneration. However, from other data we know that axon loss would likely need to be very extensive to manifest itself in functional vision loss in our behavioral assay, so milder axon loss contributing some noise to the measures cannot be excluded. But, the point raised is heard, and now we include a sentence in the discussion acknowledging that some of the signal outside of axons could have been due to degenerating axons, but still contend that our documentation of shedding intermediates support the view that many of the axonal mitochondria outside of axons were shed from otherwise intact axons.

In OPTN/MITO/LC3b trafficking experiments, does flux/number of events change? Representative kymograph in Figure 2D seems to show far more OPTN-positive mitochondria which is opposite of what is shown in Figure 2C.

Multiple reviewers rightfully point out that we did not carry out the flux experiments which would be necessary to make definitive statements regarding the amount of mitophagy. New experiments show that inhibiting lysosomal activity through chloroquine does increase the amount of astrocytic autophagosomes not yet acidified as expected, and that they contain axonal mitochondria signal, supporting the idea that astrocytes are involved in the degradation of axonal mitochondria. However, they did not show changes in the amount of stopped mitochondria, supporting the view that the co-localization of OPTN and mitochondria in axons is not conventional autophagy. This is a very important point that affects the interpretation of our results, and we thank reviewers for suggesting this experiment.

Demonstrate that axonal width measured with LC3B is representative of axonal fill/membrane marker in wt and E50K. Axonal area appears to change, is this accurate? This appears to be the case for both figure 3 and figure 4.

Addressed above.

Raw images in addition to the reconstruction would be beneficial.

Now include raw images beside the reconstruction at the first use of reconstructions.

Further characterization of exopher-like structures.

Addressed above.

Referees cross-commentingI agree with the concerns of the other reviewers, and perhaps was over-optimistic about a timeline for revision. However, I do think the work is worth the effort, and I hope to see a revised manuscript published somewhere, as these observations are novelReviewer #1 (Significance (Required)):This work reports potentially novel biology, and thus will be of interest to the field. The strength of the study is that it is an initial description of this biology, rather than a complete analysis. The work raises many more questions than it answers, and much further work on this topic is required to support these initial findings, but the manuscript will likely be of interest to many. Revisions are required to improve the rigor and clarity of the work, but following these revisions we recommend publication to facilitate follow-up work.

Fully agree that our study raises far more questions than it answers. Believe that the revisions made to address reviewer comments go a long way to improve rigor and clarity of the work. We hope that the reviewers agree and deem the changes sufficient.

Reviewer #2 (Evidence, reproducibility and clarity (Required)):Summary:This article studied transmitophagy in Xenopus optic nerves in the context of overexpressing glaucoma-associated optineurin mutations. Using a series of labeling, imaging and transplantation techniques, the authors found that overexpressing mutated optineurins stops mitochondria movements and potentially induces transmitophagy, and that astrocytes are responsible for taking up the extra-axonal mitochondria. Below are my comments on this article.Major comments:1. Identifying extra-axonal mitochondria is key to this research. In Figure 3, the authors used EGFP-LC3B as a marker for RGC boundaries. However, it is unconvincing how perfect LC3B is as a cell membrane marker. Particularly in the case of OPTN E50K OE, it seems that the optic nerve is thinner than the WT condition, which makes the quantification of extra-axonal OPTN less convincing. The authors should detect extra-axonal mitochondria with an RGC membrane marker or cytosolic marker. In addition, in Figure 3, the extra-axonal mitochondria seem to localize mostly on the dorsal surface. Why is there such a polarity?

As stated above, we acknowledge that the use of LC3b as both an autophagosome marker and a cell fill was somewhat problematic and now provide additional experiments ruling out that the LC3b expression or axon thickness in our sparse axon labeling experiments, or that E50K might affect the thickness of the optic nerve. In addition, we also provide additional new data using a *bona* fide membrane marker together a transgenic labeling or RGC mitochondria that also shows under the “baseline state” extensive mitochondria signal outside the axons on the surface of the optic nerve (New Figure 6A and new Suppl Figure 3D). All the new data are consistent with the previous data and support the view that using LC3b potentially could have been problematic, for the reasons reviewers state, but in practice it was not.

The reviewer observes that the E50K optic nerve appears thinner--this observation is not a consistent difference in optic nerves across the experimental groups. The images we show are always near the mean values for the quantitative results presented, and we rather not include prettier nerves that are not representative of the whole datasets.

As for why the extra-axonal mitochondria localize mostly to the dorsal surface, it remains undetermined.

There are dorsoventral differences in the optic nerve established during development, as developmental Sonic hedgehog signaling emanating from the midline appears to affect dorsoventral aspects of the optic nerve differentially. Early axon loss in humans and some models of glaucoma do show a dorsal bias, and there may be optic nerve lymphatic structure reported in mice that also may be preferentially dorsal. However, it is not known whether any of these observations are connected, so we did not want to speculate beyond what the data say. We do now explicitly mention the dorsoventral difference in the discussion, and state why we think it may be worth further study.

2. The experiment in Figure 5 is very important as it gives direct evidence of transmitophagy. However, one caveat is that the mitotracker injection is done after the transplantation. If in rare cases the dye is leaky after injection and is taken up by astrocytes directly, then the conclusion that mitochondria from RGCs are phagocytosed by astrocytes will be flawed. The authors should either use a transgene in the donor to label mitochondria or inject mitotracker into the donor before the transplantation and repeat the experiments. In addition, in Figure 5E, what is the large membranous structure inside the highlighted astrocyte? Is it associated with phagocytosis?

We fully agree that MitoTracker is an imperfect tool, both for the reason stated here that the dye may get into the astrocytes directly (or may label astrocyte mitochondria after it is released from degrading RGC mitochondria), and, also as stated by reviewer 3, that it requires healthy mitochondria for labeling. For this reason, we have added new datasets that rely on RGC mitochondria labeling not by Mitotracker but through a genetic reporter. As to identity of the conspicuous structure shown inside the astrocytes, it remains an open question, and we are avidly pursuing what astrocytic organelles are involved through additional transgenic reporters and correlated-light-EM studies, but those are complicated experiments that are beyond the scope of the current manuscript.

3. This research is entirely based on overexpression of OPTN. Since overexpressing WT OPTN does seem to affect mito trafficking (Figure S2G, and the description in the manuscript is often inconsistent with this result), it is unclear what the increased stalled mitochondria really mean when overexpressing mutated OPTN. Similarly, the authors examined extra-axonal mitochondria in Figures 3 and 4 all in overexpressing conditions, and made the connection that increased stalled mitochondria lead to transmitophagy. However, this conclusion will be better supported by using mutant animals rather than overexpression. The authors should consider using OPTN mutant Xenopus if available or using CRISPR to introduce the specific mutations and repeat mitochondria trafficking and transmitophagy.

We thank this reviewer by pointing out an important detail that we failed to highlight, namely that transgenic overexpression of Wt OPTN (and/or Wt LC3B) does have a small but significant effect on mitochondria trafficking. Interestingly, it is affecting just the speed of retrogradely transported mitochondria, which based on the elegant work of Holzbaur and colleagues, include mitochondria destined for degradation. So, we now acknowledge more explicitly that, since our studies involve expression of OPTN and LC3b transgenes (fluorophore tagged human genes, no less), that some caution should be exercised in not overinterpreting the results. Nonetheless, since we show that expression of Wt OPTN behaves similarly to expression of a mitochondria reporter (Tom20-mCherry) in not affecting either stopped mitochondria or extra-axonal mitochondria, we believe that our results still stand. Nonetheless, we now make mention of the effect Wt OPTN on retrograde mitochondria movement. We have embarked on OPTN loss-of-function studies and have some founder animals carrying CRISPR-generated mutations; however, these experiments will take additional time, and based on the results in mammals may or may not show any measurable effects in our assays, not only because of possible redundancy by the other damaged mitochondria adaptors that we mention in the introduction, but also because the mutations that affect the shedding process (as well as cause glaucoma) are thought to be gain-of-function mutations. However, we decided not to dwell on these complexities in the discussion, as the discussion was previously quite extensive and now is even more so with the added discussion on how our studies relate to those of exophers.

4. On Page 12, the authors claim that even overexpressing WT OPTN causes extra-axonal mitochondria in the optic nerve. However, there is no control condition without OE to support this conclusion. It is thus unclear to what extent extra-axonal mitochondria occur at baseline and how many extra-axonal mitochondria can be induced by overexpression. The authors should include, in Figure 3 and 4, controls without overexpression.

We acknowledge that our language was confusing and somewhat misleading on this point. With the caveat mentioned above that WT OPTN expression does perturb the system somewhat (by increasing the speed of mitochondria retrograde transport, perhaps by increasing the proportion of retrograde moving mitochondria tagged for degradation), we still contend that the state observed after WT OPTN expression is close to the “baseline” state. In support of that, in the new data included in response to the LC3b concern, we observe plentiful shedding events in the absence of any OPTN or LC3b transgenes. Indeed, what may be the most surprising finding of our studies is that in the absence of any significant perturbation of OPTN, there is already a large fraction of axonal mitochondria that are outside of axons and inside of astrocytes, which is consistent with what we previously observed in the optic nerve head of mice; however, the current studies provide much more rigorous quantification of the process and live imaging of intermediates, but also provide for an intervention that increases the process. While there are many more questions to answer, we do believe our studies contribute mechanistic insights.

5. A technical question regarding kymographs: Based on Figure 2C, it looks that OPTN and LC3B labeling are pretty diffuse in axons and this makes sense since they may only be associated with damaged mitos. But this raises a question about how accurate the kymograph assay is. It may significantly underestimate the fraction of OPTN/LC3B that is stationary since they appeared diffusedon the kymograph. This may explain why the percentage of stationary OPTN/LC3B is so small when the authors OE WT OPTN in Figure 2E and 2E', compared to the percentage of moving mitochondria shown in Figure 1E.

We fully agree that the kymograph studies likely underestimate the amounts of stationary mitochondria for the reasons stated. However, we interpret the discrepancy between Figure 1E and 2E and 2E’ differently. We believe that the value of stopped mitochondria in the sparse labeling experiments are actually more accurate, as the value of stopped mitochondria in the whole nerve experiments likely include mitochondria stopped within the axons, but also mitochondria recently shed either by those or nearby axons which are perceived to be in axons due to limitations of imaging resolution. In the discussion we now make very explicit that all the measures we provide need should be interpreted as estimates, as every experiment relies on assumptions and is subject to technical limitations.

Minor:1. Figure 2E and 2E' do not agree with the text on page 7 and page 8. Not only F178A, but also H486R and D474N have no effect on OPTN trafficking. The authors should make their conclusions more accurate.

F178 was the only mutation that had no effect on either OPTN or LC3b in either F_0_ or F1 experiments. However, we agree that our language should have been clearer, and now we have made our description of the results (and conclusions) more accurate.

2. Figure S2E-F: why does OE of mutated OPTN in F1s but not in F0s reduce trafficking speed compared to WT?

We do not know the reason for this discrepancy. Though it does not wholly agree with the rest of the story, we felt it important to include all relevant data, not only that which perfectly fit our interpretation. One possible reason may be that the F1 data derives from a single integration event, which is the reason why we trust more the F_0_ data that derive from multiple integrations, in what are essentially outbred animals, which is the reason we present the F_0_ data as the primary results where possible.

3. In movie 5, fusion of exopher with other structures is not clear and also the GFP signal does not disappear, which is in contrast to the statement in the text that the GFP signal is quenched in acidified environment. To confirm that LC3B leaves RGC axons in exophers, the authors should consider switching the fluorophores and examine LC3B localization during exopher formation.

This too is a valid point, and we have amended our description of these results. While swapping fluorophores between OPTN and LC3b is a highly worthy experiment, for technical reasons it likely would take many months to carry out just because of how involved it is to make the relevant constructs (recombineering details provided in the methods section).

4. In figure 6, to better show exopher formation and the pinching-off step, the authors should consider labeling the membrane and mitochondria instead of using the LC3B and OPTN marker.

This arguably was the biggest weakness of our initial submission, and now provide new experiments using a *bona fide* membrane marker. We have not yet captured a pinching-off event with these better reporters, but that is not surprising given how rare they are, which we now quantify. Indeed, a membrane reporter and a mitochondria transgene in sparsely labeled axons are the ideal tool for figuring out the frequency of these structures and what fraction contain mitochondria, data which we now provide.

Referees cross-commentingGenerally agree with the criticisms voiced by the other reviewers; in aggregate the reviews indicate the manuscript needs more than just a quick fix.Reviewer #2 (Significance (Required)):Previous literature has already described the transmitophagy process in the optic nerve. The significance of this paper lies in the observation that overexpressing glaucoma-associated OPTN mutants can induce increased transmitophagy through astrocytes, which points to a potential role of OPTN in glaucoma. A highlight of this paper is the use of correlated light SBEM to directly show transmitophagy in astrocytes. However, the significance of this paper may be limited for the following reasons: 1. everything is based on overexpression of mutated OPTN, which makes it hard to translate the results to real disease conditions; 2. The consequence of increased transmitophagy on RGC survival or visual functions is unclear.

While we agree that much of the paper is based on OPTN overexpression, we did have experiments and now provide more that were not based on OPTN overexpression. Some of these still involve expression of a different transgene (Tom20-mCherry) that might in principle perturb the system, though we show that expression of Tom20-mCherry does not affect mitochondria movement parameters as measured by Mitotracker. As to “the consequence of increased transmitophagy”, we do now provide data showing that there is no vision loss suggestive of axon loss or severe dysfunction at the time that the imaging studies were carried out. Whether longer term expression of these OPTN transgenes lead to axon degeneration and visual dysfunction are studies that are ongoing, but those studies involve extensive characterizations and controls that are beyond what could be included in this study.

Reviewer #3 (Evidence, reproducibility and clarity (Required)):SummaryIn this work, Jeong et al. describe the effect of Optineurin (OPTN) mutations in the transcellular degradation of retinal ganglion cell (RGC) mitochondria by astrocytes at the Optic Nerve (ON), a process previously described this group and referred as "transmitophagy" (Davis et al. 2014). Here, authors use *Xenopus laevis* animal model to image the optic nerve of animals carrying different OPTN mutations associated to disease or with compromised function and explore its effect in mitochondria dynamics at the RGC axons. They find that OPTN mutants lead to increased stationary mitochondria in the nerve and affect their co-localization with mitophagy-related markers, suggesting alterations in this pathway. Finally, they found that mitochondria colocalizing with OPTN can be found in the periphery of the ON under different conditions and this is particularly increased in glaucoma-associated E50K mutation. This extracellular mitochondria are transferred in vesicles to astrocytes, as they previously described in mice (Davis 2014), where they are presumably degraded.Major comments:OPTN levels at a given time point cannot be used as readout for mitophagy level/flux. Both OPTN and LC3b are degraded upon fusion with acidic compartment (i.e. lysosomes, PMID: 33783320, 33634751) and that is the reason why the field of autophagy /mitophagy blocks lysosomal activity to measure autophagy/mitophagy flux (PMID: 33634751). In this document, authors claim that there is low levels of mitophagy in RGC axons at baseline and increased levels of mitophagy in glaucoma associated perturbations just based on increased presence of OPTN+ mitochondria in this condition. This could be also interpreted as an accumulation of nondegraded defective mitochondria due to a mitophagy block in neurons carrying the glaucoma associated mutation, which is the opposite of what they propose. If authors want to evaluate mitophagy levels in this system, mitophagy/autophagy flux experiments should be performed.

In response to reviewers, we do now include “lysosome inhibition” experiment, using chloroquine at doses modestly above those used in aquaculture as an anti-parasitic. After testing various chemical means to inhibit lysosome activity, it was the only one that did not adversely affect the animals. We know the chloroquine intervention works because we see the expected increase in autophagosomes using the standard LC3btandem reporter, and in those unacidified astrocytic autophagosomes we do indeed find axonal mitochondria signal. However, since the amount of mitochondria signal there is small relative to the total amount of axonal mitochondria in the astrocytes, we do not feel it would be appropriate to make mechanistic claims, for example claiming this to be related to LC3b associated phagocytosis; much more work would be needed to make that claim. However, we were surprised to find no alteration in either stopped mitochondria in axons or axonal mitochondria material within the astrocytes. There are technical reasons why this result might be difficult to interpret, but now having done it (as we should have before), we are even more careful in describing the process as transcellular degradation rather than transmitophagy. We elaborate further on this point in the next response.

I find inappropriate the use of the term "transmitophagy". Although this term transmits very well the message that the authors try to strength, the term "mitophagy" refers to the specific elimination of mitochondria through autophagy (PMID: 21179058). There are many reasons why I think that "transmitophagy" is not adequate to describe this phenomena but I will just refer to these three: First, authors do not provide data showing that this mechanism is specific for mitochondria as they have never checked for the presence of other type of cargo in the vesicles produced by RGCs. If these are related to exophers as they suggest in the document, is very probable that they contain other type of cargo; Second, if the final destiny for those particles is the acidic compartment of astrocytes, this process may have nothing to do with autophagy/mitophagy and just share some molecular mediators with those pathways; Third, they should explore if other canonical mitophagy molecular mediators (i.e. Parkin/Pink) are regulating the production or the mitochondria recruitment to this extracellular particles.

We too struggle with our own “transmitophagy” term, for the very reasons stated. To address this concern, we now refer to the process as “transcellular degradation of mitochondria”, which is how we described it initially in mice as well. We do present new data that show that while the majority of axonal outpocketings contain mitochondria, not all do. This suggests that the others may contain other cargo, which supports the view that what we are dealing with in axons are indeed exophers. And yet, since what we measure is mitochondria, we think most appropriate to describe the process narrowly and not extrapolate to other types of exophers. We agree that what we originally discovered in mice and now live image and perturb in frog, may not be “autophagy” according to the strict definition of the term, but rather a process that uses some of the same molecular machinery, which given the evolutionary link between autophagy and phagocytosis that should be no surprise. Terminology can be tricky, and we thank the reviewer for calling us out on this point. We now use the term “transmitophagy” only once in the Discussion section making the link between our work and the emerging field of exopher biology, and use that occasion to elaborate the point that the more descriptive term “transcellular degradation of mitochondria” is more appropriate in our case.

In several experiments, authors use Mitotracker instead of genetic tools to quantify the amount of mitochondria co-localizing with OPTN (Figure 2, Figure 3) or being transferred to astrocytes (Figure 4). A problem here is that Mitotracker needs the mitochondria to be active at the time of injection in order to label them (PMID: 21807856) and it has a clear effect in mitochondria dynamics in their setting, as pointed by the authors. Since most mitochondria transferred to astrocytes would be presumably damaged and not able to import Mitotracker, I am concern about how this is affecting their quantifications and the conclusions.

We agree. The use of Mitotracker to label the RGC mitochondria can be problematic for the reasons stated by reviewers 1 and 3. Indeed, our opinion is that many of the studies out there that claim to demonstrate transfer of mitochondria between cells likely are just showing the transfer of the dye rather than the mitochondria. While the previous submission included a number of controls to address this concern, we now provide multiple new experiments that measure the transfer of mitochondria through a transgene rather than Mitotracker. The provided experiments use a new Tom20-mCherry transgene which is highly specific to mitochondria due to the use of an SOD2 UTR. We have similar data using RGC-expressed Mito-mCherry and Mito-EGFP-mCherry (using the commonly used Cox8 mitochondria matrix targeting sequence); we do not include such data because we find the provided data sufficiently compelling, and the story is already sufficiently long and complicated.

Some conclusions are based on single images with no quantifications or statistics. This is the case for: Page 6 "Most of the mCherry and Mitotracker objects colocalized with each other both in the merged images (Figure S1C) and kymographs (Figure S1D), indicating that the mitochondria-targeted transgene and Mitotracker similarly label the RGC axonal mitochondria".

That is a fair comment. After reanalyzing the original dataset used, it would be very difficult to quantify that statement, largely because the Tom20-mCherry expression was relatively weak in those particular animals. We are confident that we could generate a new dataset to provide support for this statement, but instead chose to just provide side-by-side movies of mitochondria labeled by Mitotracker or the Tom20-mCherry transgenes, which we believe is far more compelling than any quantification we could provide.

Page 8 "In the nerves labeled by Mitotracker, visual inspection of the raw images (Figure 2C) and the derived kymographs (Figure 2D) showed that OPTN and the Mitotracker labeled mitochondria often co-localized, particularly in the stopped populations, and more so in the animals expressing E50K OPTN, further suggesting that at least a fraction of the stopped LC3b, OPTN and mitochondria might represent mitophagy occurring in the axons".

While we have made a minor change to this sentence, we feel that it is appropriate given that it serves just as a justification to carry out the quantitative studies that follow. We would not have quantified the process had it not been obvious to the eye. However, we do not interpret the results as supporting that mitophagy occurs in axons, for the reasons explained above.

Page 14 "We also observed similar axonal dystrophies and exopher-like structures in E50K OPTN under similar imaging settings, but with 2-min intervals and additional Mitotracker labeling (Video 6), demonstrating that these structures not only contain OPTN but also mitochondria or mitochondria remnants". Image in video is not clear and there is not quantification for OPTN or OPTN+ mitochondria.

We have removed Video 6.

In Figures showing the reconstruction of OPTN+ mitochondria outside nerve (Figure 3 and Figure 4), those seem to be present only in one lateral of the nerve. Is this process polarized in any way (i.e. faced to astrocytes) or is the result of a technical issue (i.e. difference in laser penetration for blue vs Yellow lasers)? I think it will be important to include this in the discussion.

This was also pointed out by reviewer 1, and we agree that it is worth including in the discussion, which we now do. While we do not believe it to be a light penetration issue (based on fluorescence intensities and apparent spatial resolution), we also do not yet have an explanation. Having studied dorsoventral differences in the visual pathway both during my graduate and post-doctoral years, I am very interested in this asymmetry, and we have some theories that might explain it, mentioned above. The asymmetry is obvious and thus we think it would have been inappropriate not to show, but it also be inappropriate to be overly speculative.

In Pag.13 authors claim "OPTN and mitochondria leave RGC axons in the form of exophers". After "exophers" were coined by the Driscoll lab in 2017, too few people has adopted this terminology and the molecular machinery involved in this process is still under research. It is clear that the particles described here share some similarities with exophers like size (in the range of microns) and cargo (mitochondria), but you have not demonstrated if they share the same origin or are part of the same phenomena. For that reason, I recommend to be more cautious with this statement and point these limitations in the discussion. Additionally, since Exophers are not a consensus or well defined particles, authors should include an introductory paragraph at the beginning of this section for readers to understand what they are talking about.

We wholly agree with all points. We now have moved all mention of exophers to just the discussion.

Exophers described by Monica Driscoll and Andres Hidalgo laboratories are presented as "garbage bags" that help cells to stay fit through elimination of unwanted material. If the extracellular vesicles presented here are part of the same mechanism and potentially beneficial for the RGCs, why are they increased in OPTN mutants? Is it part of RGCs response to a proteomic stress generated by malfunctioning OPTN? I think that is critical to understand this to figure out the relevance of your findings.

Our personal opinion is that the OPTN mutants most likely lead to stress focally in the axons, thus triggering exopher generation. We are carrying additional experiments to determine whether too much exopher generation or their insufficient degradation by astrocytes might be deleterious (by causing inflammation). However, those are big stories that would not stand on their own were we not able to first rigorously demonstrate that certain OPTN mutants increase exopher generation, which I believe our study demonstrates, albeit now without calling them exophers.

Related to Figure 5G, authors say "The soma of the astrocytes were located at the optic nerve periphery but had processes that extended deep into the parenchyma". This is very interesting and opens the possibility that many mitochondria are directly transferred to astrocytes through that processes instead of the lateral of the nerve, meaning that your quantifications of "transmitophagy" may be underestimated.

We also agree that this. Our limited optical resolution, and limitations intrinsic to carrying out quantifications with Imaris software, are likely the main reasons for the discrepancy between the whole nerve and sparselabelled-axon estimates of how much axonal material is outside of axons. Our view is that most of the transcellular degradation occurs within fine astrocyte processes, and that only in the case of failure to degrade material in these fine processes that significant amounts accumulate in the cell body (optic nerve periphery), and that in the cell body additional or different degradative pathways are utilized. Experiments using various transgenes and correlated EM as well as perturbation experiments are ongoing attempting to firmly establish what organelles are used in processes versus soma. However, we believe that such studies are well beyond the scope of this manuscript..

Reference to Figure S2G is missing.

Now mentioned twice. Thank you.

I cannot find in Figure 5 E-I legends what are the cells/structures labelled in Green and Red.

Thank you.

Referees cross-commentingIn agreement with my colleagues, I think that a revision is needed to support some important points of the paper. The work is interesting and I think it deserves a chance for revision. Having that said, I am not familiar with the breeding and experimental times when working with Xenopus but, considering the amount of work requested, it may require more than 3 months to have the work done.Reviewer #3 (Significance (Required)):Until not very long ago, it was thought that mitochondria could not cross cell barriers. In recent years however, there has been an explosion in the number of works showing mitochondria transfer between different cell types in vivo. This may happen either as an organelle donation to improve energy production or as a quality control mechanism to get rid of damaged mitochondria, as it is the case in this work. The laboratory of Nicholas Marsh-Armstrong was pioneer in this field with a foundational work in 2014 where they show how RGC-derived mitochondria are captured and eliminated by astrocytes in mice (PMID: 24979790). This work was particularly relevant because it proposed for the first time that mitochondrial degradation can occur in RGC axons far from the cell soma, and surrogated in a different cell type, something that changed completely the view of how quality control is maintained in neurons and other cell types.In the present study, Jeong and collaborators explore how Glaucoma-associated Optineurin mutations affect this process, which is of potential interest for the broad cell biologist community due to its possible implications in other tissues and cell types (OPTN is broadly expressed), but especially for those researchers interested in neurobiology, quality control mechanisms and mitochondria biology. Since some OPTN mutations studied here cause disease, they are also relevant for the clinic.This work provides a thorough characterization of how relevant Optineurin mutations affect mitochondria dynamics in RGCs and their transference to astrocytes, as fairly claimed in the title. However, the mechanism by which they result in pathology is not either explored or carefully discussed, making this a descriptive work with no much conceptual insight. In addition, conclusions are often not unambiguously stated and the results part contains a lot of large sentences and unnecessary technical data that hinders reading and difficult the transmission of the key messages.Even if it stands as a descriptive work, the physiological and clinical relevance of these findings is not clear.There are some claims related with mitophagy activity that may require more sophisticated experiments (mitophagy flux with lysosomal inhibitors). Please see comments above. A critical point to understand the relevance of this work would be to demonstrate if alterations in transmitophagy are either causing or involved in the disease generated by these OPTN mutations in any way, or just a correlative phenomenon. To help authors contextualize my point of view, my field of expertise includes cell biology, imaging, quality control pathways, mitochondria biology and phagocytosis, among others. I am not familiar with *Xenopus laevis* genetics or the limitations to work with this animal model.

We appreciate both the complements and the critiques. To a fault, we rather undersell than oversell. We are actively pursuing the possibility that dysregulation of this process is disease causing, and not just for glaucoma. However, those studies will not stand without a strong foundation, which we believe this study provides.